# Curriculum Reinforcement Learning using Optimal Transport via Gradual Domain Adaptation

**Peide Huang, Mengdi Xu, Jiacheng Zhu, Laixi Shi, Fei Fang, Ding Zhao**
Carnegie Mellon University
Pittsburgh, PA 15213
{peideh, mengdixu, jzhu4, laixis, feifang, dingzhao}@andrew.cmu.edu

## Abstract

Curriculum Reinforcement Learning (CRL) aims to create a sequence of tasks, starting from easy ones and gradually learning towards difficult tasks. In this work, we focus on the idea of framing CRL as interpolations between a *source* (auxiliary) and a *target* task distribution. Although existing studies have shown the great potential of this idea, it remains unclear how to formally quantify and generate the movement between task distributions. Inspired by the insights from gradual domain adaptation in semi-supervised learning, we create a natural curriculum by breaking down the potentially large task distributional shift in CRL into smaller shifts. We propose GRADIENT, which formulates CRL as an optimal transport problem with a tailored distance metric between tasks. Specifically, we generate a sequence of task distributions as a geodesic interpolation (i.e., Wasserstein barycenter) between the source and target distributions. Different from many existing methods, our algorithm considers a task-dependent contextual distance metric and is capable of handling nonparametric distributions in both continuous and discrete context settings. In addition, we theoretically show that GRADIENT enables smooth transfer between subsequent stages in the curriculum under certain conditions. We conduct extensive experiments in locomotion and manipulation tasks and show that our proposed GRADIENT achieves higher performance than baselines in terms of learning efficiency and asymptotic performance.

## 1 Introduction

Reinforcement Learning (RL) [1] has demonstrated great potential in solving complex decision-making tasks [2], including but not limited to video games [3], chess [4], and robotic manipulation [5]. Among them, various prior works highlight daunting challenges resulting from sparse rewards. For example, in the maze navigation task, since the agent needs to navigate from the initial position to the goal to receive a positive reward, the task requires a large amount of randomized exploration. One solution to address this issue is Curriculum Reinforcement Learning (CRL) [6, 7], of which the objective is to create a sequence of environments to facilitate the learning of difficult tasks.

Although there are different interpretations of CRL, we focus on the one that views a curriculum as a sequence of task distributions that interpolate between a *source* (auxiliary) task distribution and a *target* task distribution [8, 9, 10, 11, 12, 13]. This interpretation allows more general presentations of tasks and a wider range of objectives such as generalization (uniform distribution over the task collection) or learning to complete difficult tasks (a subset of the task collection). Again we use the maze navigation as an example. Given a fixed maze layout and goal, a task is then defined by a start position, and the task distribution is a categorical distribution over all possible start positions. With a target distribution putting mass over start positions far away from the goal position, a natural

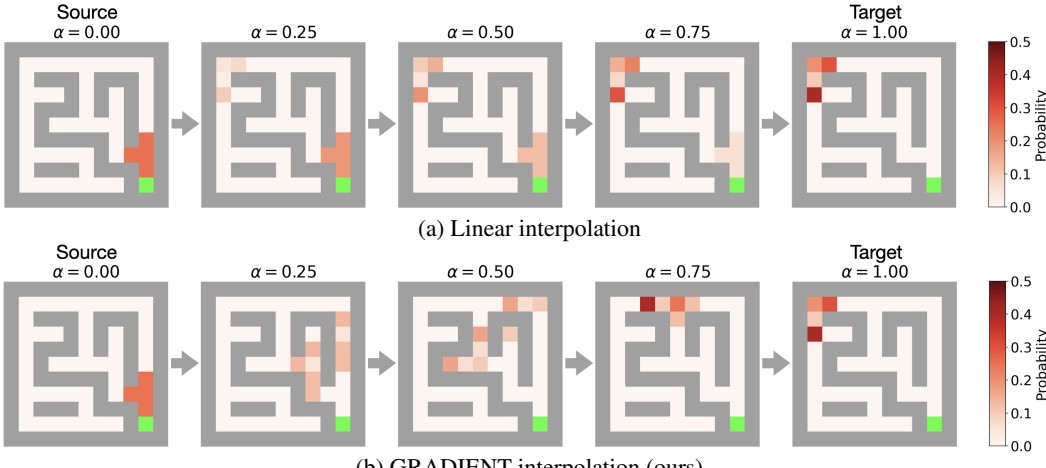

Figure 1: Intermediate task distributions generated by (a) linear interpolation and (b) our method for the maze navigation task. The green cell represents the goal. The red cells represent the initial positions (the darker the color is, the higher the probability is). The first column shows the source task distribution and the last column shows the target task distribution. In (a), linear interpolations do not cover cells where the source and the target have zero probability, which hardly benefits the learning. In contrast, (b) GRADIENT interpolations creates a curriculum that gradually morphs from the source to the target, covering tasks of intermediate difficulty and improving the learning efficiency.

curriculum to accelerate the learning process is putting start positions close to the goal position first, and gradually moving them towards the target distribution.

However, most of the existing methods, that interpret the curriculum as shifting distributions, use Kullback–Leibler (KL) divergence to measure the distance between distributions. This setting imposes several restrictions. First, due to either problem formulations or the computational feasibility, existing methods often require the distribution to be parameterized, e.g., Gaussian [8, 9, 10, 13], which limits the usage in practice. Second, most of the existing algorithms using KL divergence implicitly assume an $l_2$ Euclidean space which ignores the manifold structure when parameterizing RL environments [14].

In light of the aforementioned issues with the existing CRL method, we propose GRADIENT, an algorithm that creates a sequence of task distributions gradually morphing from the source to the target distribution using optimal transport (OT). GRADIENT approaches CRL from a gradual domain adaptation (GDA) perspective, breaking the potentially large domain shift between the source and the target into smaller shifts to enable efficient and smooth policy transfer. In this work, we first define a distance metric between individual tasks. Then we can find a series of task distributions that interpolate between the easy and the difficult task distribution by computing the Wasserstein barycenter. GRADIENT is able to deal with both discrete and continuous environment parameter spaces, and nonparametric distributions (represented either by explicit categorical distributions or implicit empirical distributions of particles). Under some conditions [15], GRADIENT provably ensures a smooth adaptation from one stage to the next.

We summarize our main contributions as follows:

1. We propose GRADIENT, a novel CRL framework based on optimal transport to generate gradually morphing intermediate task distributions. As a result, GRADIENT requires little effort to transfer between subsequent stages and therefore improves the learning efficiency towards difficult tasks.

2. We develop $\pi$-contextual-distance to measure the task similarity and compute the Wasserstein barycenters as intermediate task distributions. Our proposed method is able to deal with both continuous and discrete context spaces as well nonparametric distributions. We also prove the theoretical bound of policy transfer performance which leads to practical insights.

3. We demonstrate empirically that GRADIENT has stronger learning efficiency and asymptotic performance in a wide range of locomotion and manipulation tasks when compared with state-of-the-art CRL baselines.

## 2 Related Work

**Curriculum reinforcement learning.**   Curriculum reinforcement learning (CRL) [6, 16] focuses on the generation of training environments for RL agents. There are several objectives in CRL: improving learning efficiency towards difficult tasks (*time-to-threshold*), maximum return (*asymptotic performance*), or transfer policies to solve unseen tasks (*generalization*). From a domain randomization perspective, Active Domain Randomization [5, 17] uses curricula to diversify the physical parameters of the simulator to facilitate the generalization in sim-to-real transfer. From a game-theoretical perspective, adversarial training is also developed to improve the robustness of RL agents in unseen environments [18, 19, 20, 21]. From an intrinsic motivation perspective, methods have been proposed to create curricula even in the absence of a target task to be accomplished [22, 13, 23].

**CRL as an interpolation of distributions.**   In this work, we focus on another stream of works that interprets CRL as an explicit interpolation between an auxiliary task distribution and a difficult task distribution [8, 9, 10, 11]. Self-Paced Reinforcement Learning (SPRL) [8] is proposed to generate intermediate distributions by measuring the task distribution similarity using Kullback–Leibler (KL) divergence. However, as we will show in this paper, KL divergence brings several shortcomings that may impede the usage of the algorithms. First, although the formulation of [8, 9, 10, 11] does not restrict the distribution class, the algorithm realization requires the explicit computation of KL divergence, which is analytically tractable only for a restricted family of distributions. Second, using KL divergence implicitly assumes a $l_2$ Euclidean space which ignores the manifold structure when parameterizing RL environments. In this work, we use Wasserstein distance instead of KL divergence to measure the distance between distributions. Unlike KL divergence, Wasserstein distance considers the ground metric information and opens up a wide variety of task distance measures.

**CRL using Optimal Transport.**   Hindsight Goal Generation (HGG) [24] aims to solve the poor exploration problem in Hindsight Experience Replay (HER). HGG computes 2-Wasserstein barycenter approximately to guide the hindsight goals towards the target distribution in an implicit curriculum. Concurrent to our work, CURROT [25] also uses optimal transport to generate intermediate tasks explicitly. CURROT formulates CRL as a constrained optimization problem with 2-Wasserstein distance to measure the distance between distributions. The main difference is that we propose task-dependent contextual distance metrics and directly treat the interpolation as the geodesic from the source to the target distribution.

**Gradual domain adaptation in semi-supervised learning**   Gradual domain adaptation (GDA) [26, 27, 28, 29, 30, 31, 32, 33] considers the problem of transferring a classifier trained in a source domain with labeled data, to a target domain with unlabelled data. GDA solves this problem by designing a sequence of learning tasks. The classifier is retrained with the pseudolabels created by the classifier from the last stage in the sequence. Most of the existing literature assumes that there exist intermediate domains. However, there are a few works aiming to tackle the problem when intermediate domains, or the index (i.e., stage in the curriculum), are not readily available. A coarse-to-fine framework is proposed to sort and index intermediate domain data [33]. Another study proposes to create virtual samples from intermediate distributions by interpolating representations of examples from source and target domains and suggests using the optimal transport map to create interpolated data in semi-supervised learning [32]. It is demonstrated theoretically in [27] that the optimal path of samples is the geodesic interpolation defined by the optimal transport map. Our work is inspired by the *divide and conquer* paradigm in GDA and also uses the geodesic as our curriculum plan (although in a different learning paradigm).

## 3 Preliminary

### 3.1 Contextual Markov Decision Process

A contextual Markov decision process (CMDP) extends the standard single-task MDP to a multi-task setting. In this work, we consider discounted infinite-horizon CMDPs, represented by a tuple $M = (\mathcal{S}, \mathcal{C}, \mathcal{A}, R, P, p_0, \rho, \gamma)$. Here, $\mathcal{S}$ is the state space, $\mathcal{C}$ is the context space, $\mathcal{A}$ is the action space, $R : \mathcal{S} \times \mathcal{A} \times \mathcal{C} \mapsto \mathbb{R}$ is the context-dependent reward function, $P : \mathcal{S} \times \mathcal{A} \times \mathcal{C} \mapsto \Delta(\mathcal{S})$ is the context-dependent transition function, $p_0 : \mathcal{C} \mapsto \Delta(\mathcal{S})$ is the context-dependent initial state distribution, $\rho \in \Delta(\mathcal{C})$ is the context distribution and $\gamma \in (0, 1)$ is the discount factor. Note that goal-conditioned reinforcement learning [12] can be considered as a special case of the CMDP.

To sample a trajectory $\tau := \{s_t, a_t, r_t\}_{t=0}^{\infty}$ in CMDPs, the context $c \sim \rho$ is randomly generated by the environment at the beginning of each episode. With the initial state $s_0 \sim p_0(\cdot \mid c)$, at each time step $t$, the agent follows a policy $\pi$ to select an action $a_t \sim \pi(s_t, c)$ and receives a reward $R(s_t, a_t, c)$. Then the environment transits to the next state $s_{t+1} \sim P(\cdot \mid s_t, a_t, c)$. Contextual reinforcement learning naturally extends the original RL objective to include the context distribution $\rho$. To find the optimal policy, we need to solve the following optimization problem:

$$\max_{\pi} V^{\pi}(\rho) = \max_{\pi} \mathbb{E}_{\tau} \left[ \sum_{t=0}^{\infty} \gamma^t R(s_t, a_t, c) \mid c \sim \rho; \pi \right] \tag{1}$$

### 3.2 Optimal Transport

**Wasserstein distance.** The Kantorovich problem [34], a classic problem in optimal transport [35], aims to find the optimal coupling $\theta^*$ which minimizes the transportation cost between measures $\mu, \nu \in \mathcal{M}(\mathcal{C})$. Therefore, the *Wasserstein distance* defines the distance between probability distributions:

$$\mathcal{W}_d(\mu, \nu) = \inf_{\theta \in \Theta(\mu, \nu)} \int_{\mathcal{C} \times \mathcal{C}} d(c_s, c_t) \, \mathrm{d}\theta(c_s, c_t), \text{ subject to } \Theta = \left\{ \theta : \gamma_{\#}^{\mathcal{C}} \theta = \mu, \gamma_{\#}^{\mathcal{C}} \theta = \nu \right\} \tag{2}$$

where $\mathcal{C}$ is the support space, $\Theta(\mu, \nu)$ is the set of all couplings between $\mu$ and $\nu$, $d(\cdot, \cdot) : \mathcal{C} \times \mathcal{C} \mapsto \mathbb{R}_{\geq 0}$ is a distance function, $\gamma^{\mathcal{C}}$ is the projection from $\mathcal{C} \times \mathcal{C}$ onto $\mathcal{C}$, and $T_{\#}P$ generally denotes the push-forward measure of $P$ by a map $T$ [36, 37, 38]. This optimization is well-defined and the optimal $\theta$ always exists under mild conditions [35].

**Wasserstein Geodesic and Wasserstein Barycenter.** To construct a curriculum that allows the agent to efficiently solve the difficult task distribution, we follow the *Wasserstein geodesic* [39], the shortest path [27] under the Wasserstein distance between the source and the target distributions. While the original Wasserstein barycenter [40, 41] problem focuses on the Frèchet mean of multiple probability measures in a space endowed with the Wasserstein metric, we consider only two distributions $\mu_0$ and $\mu_1$. The set of barycenters between $\mu_0$ and $\mu_1$ is the geodesic curve given by McCann's interpolation [42, 43]. Thus, the interpolation between two given distributions $\mu_0$ and $\mu_1$ is defined as:

$$\nu_{\alpha} = \arg\min_{\nu_{\alpha}'} (1 - \alpha) \mathcal{W}_d(\mu_0, \nu_{\alpha}') + \alpha \mathcal{W}_d(\nu_{\alpha}', \mu_1), \tag{3}$$

where each $\alpha \in [0, 1]$ specifies one unique interpolation distribution on the geodesic. While the computational cost of Wasserstein distance objective in Equation (2) could be a potential obstacle, we can follow the entropic optimal transport and utilize the celebrated Sinkhorn's algorithm [44]. Moreover, we can adopt a smoothing bias to solve for scalable debiased Sinkhorn barycenters [45].

## 4 Curriculum Reinforcement Learning using Optimal Transport

We first formulate the curriculum generation as an interpolation problem between probability distributions in Section 4.1. Then we propose a distance measure between contexts in Section 4.2 in order to compute the Wasserstein barycenter. Next, we show our main algorithm, GRADIENT, in Section 4.3 and an associated theorem that provides practical insights in Section 4.4.

### 4.1 Problem Formulation

Formally, given a target task distribution $\nu(c) \in \Delta(\mathcal{C})$, we aim to automatically generate a curriculum of task distributions, $\rho_0(c), \rho_1(c), \ldots, \rho_K(c)$ with $K$ stages, that enables the agent to gradually adapt from the auxiliary task distribution $\mu(c)$ to the target $\nu(c)$, i.e., $\rho_0(c) \to \mu(c)$ and $\rho_K(c) \to \nu(c)$. If the context space $\mathcal{C}$ is discrete (sometimes called fixed-support in OT [35]) with cardinality $|\mathcal{C}|$, the task distribution is represented by a categorical distribution, e.g., $\mu(c) = [p(c_1), p(c_2), ..., p(c_{|\mathcal{C}|})]$, where $p(c_i) \geq 0, \sum_i p(c_i) = 1$. Whereas if the context space $\mathcal{C}$ is continuous (sometimes called free-support in OT), the task distribution is then approximated by a set of particles sampled from the distribution, e.g., $\mu(c) \approx \hat{\mu}(c) = \frac{1}{n_s} \sum_{i=1}^{n_s} \delta_{c_{si}}(c)$, where $\delta_{c_{si}}(c)$ is a Dirac delta at $c_{si}$. This highlights the capability of dealing with nonparametric distributions in our formulation and algorithms, in contrast to existing algorithms [8, 13, 8, 9, 10, 11].

## 4.2 Contextual Distance Metrics

To define the distance between contexts, we start by defining the distance between states. **Bisimulation metrics** [46, 47, 48] measure states' *behaviorally equivalence* by defining a recursive relationship between states. Recent literature in RL [49, 14] uses the bisimulation concept to train state encoders that facilitate multi-tasking and generalization. [50] uses this bisimulation concept to enforce the policy to visit state-action pairs close to the support of logged transitions in offline RL.

However, the bisimulation metric is inherently "pessimistic" because it considers equivalence under all actions, even including the actions that never lead to positive outcomes for the agent [48]. To address this issue, [48] proposes on-policy $\pi$-bisimulation that considers the dynamics induced by the policy $\pi$. Similarly, we extend this notion to the CMDP settings and define the **contextual $\pi$-bisimulation metric**:

$$d_{c_i,c_j}^\pi(s_i, s_j) = |R_{s_i,c_i}^\pi - R_{s_j,c_j}^\pi| + \gamma \mathcal{W}_d(\mathcal{P}_{s_i,c_i}^\pi, \mathcal{P}_{s_j,c_j}^\pi) \tag{4}$$

where $R_{s,c}^\pi := \sum_a \pi(a \mid s) R(s, a, c)$ and $P_{s,c}^\pi := \sum_a P(\cdot \mid s, a, c)\pi(a \mid s)$. With the definition of contextual $\pi$-bisimulation metric, we are ready to propose the $\pi$-**contextual-distance** to measure the distance between two contexts:

**Definition 4.1 ($\pi$-contextual-distance)** *Given a CMDP $M = (\mathcal{S}, \mathcal{C}, \mathcal{A}, R, P, p_0, \rho, \gamma)$, the distance between two contexts $d^\pi(c_i, c_j)$ under the policy $\pi$ is defined as*

$$d^\pi(c_i, c_j) = \mathbb{E}_{s_i \sim p_0(\cdot|c_i), s_j \sim p_0(\cdot|c_j)} \left[ d_{c_i,c_j}^\pi(s_i, s_j) \right] \tag{5}$$

Conceptually, the $\pi$-**contextual-distance** approximately measures the performance difference between two contexts $c_i$ and $c_j$ under $\pi$. The algorithm to compute a simplified version of this metric under some conditions is detailed in the Appendix C.1. Note that it is difficult to compute this metric precisely in general. In practice, we can design and compute some surrogate contextual distance metrics, depending on the specific tasks. There are situations where it is reasonable to use $l_2$ as a surrogate metric when the contextual distance resembles the Euclidean space, such as in some goal-conditioned continuous environments [24]. In addition, although the contextual distance is not a strict metric, we can still use it as a ground metric in the OT computation. With the concepts of a contextual distance metric, we now introduce the algorithms to generate intermediate task distribution to enable the agent to gradually transfer from the source to the target task distribution.

## 4.3 Algorithms

We present our main algorithm in Algorithm 1. We introduce an interpolation factor $\Delta\alpha$ to decide the difference between two subsequent task distributions in the curriculum. Smaller $\Delta\alpha$ means a smaller difference between subsequent task distributions. In the algorithm, we treat $\Delta\alpha$ as a constant for simplicity but in effect, it can be scheduled or even adaptive, which we leave to future work. Note that the $\Delta\alpha$ is analogical to the KL divergence constraint in [9, 10, 8]. The main difference is that we use Wasserstein distance instead of KL divergence.

At the beginning of each stage in the curriculum, we add a $\Delta\alpha$ to the previous $\alpha$ (starting from 0). Then we pass the $\alpha$ into the function `ComputeBarycenter` (Algorithm 2) to generate the intermediate task distribution. Note that when $\alpha = 0$ and 1, the generated distribution are the source and target distribution respectively. In the `ComputeBarycenter` function, the computation method differs depending on whether the context is discrete or continuous. After generating the intermediate task distribution, we optimize the agent in this task distribution until the accumulative return $G$ reaches the threshold $\bar{G}$. Then the curriculum enters the next stage and repeats the process until $\alpha = 1$. In other words, the path of the intermediate task distributions is the OT geodesic on the manifold defined by Wasserstein distance.

## 4.4 Theoretical Analysis

The proposed algorithm benefits from breaking the difficulty of learning in the target domain into multiple small challenges by designing a sequence of $K$ stages that gradually morph towards the target. This motivates us to theoretically answer the following question:

*Can we achieve smooth transfer to a new stage based on the optimal policy of the previous stage?*

**Algorithm 1:** **GRA**dual **D**omain adaptation for curriculum re**I**nforc**E**ment lear**N**ing via optimal **T**ransport (**GRADIENT**)

---

**Input:** Source task distribution $\mu(c)$, target task distribution $\nu(c)$, interpolation factor $\Delta\alpha$, distance metric $d$, reward threshold $\bar{G}$, maximum number of stages $K$.

Initialize the agent policy $\pi$
**for** $k$ *in* $0, 1, 2, ..., K$ **do**
    $\alpha \leftarrow \min(k * \Delta\alpha, 1)$;
    $\rho(c) \leftarrow$
      ComputeBarycenter$(\mu, \nu, \alpha, d)$;
    **do**
        $G \leftarrow$ Optimize $\pi$ in the task distribution $\rho(c)$ and return training reward;
    **until** $G > \bar{G}$;
**Output:** Agent policy $\pi$

---

**Algorithm 2:** ComputeBarycenter

---

**Input:** Source and target $\mu, \nu$, interpolation constant $\alpha$, distance metric $d$.
$w_1 \leftarrow \alpha, w_2 \leftarrow 1 - \alpha, (b_1, b_2), l \leftarrow \mathbf{1}$;
**if** $\mathcal{C}$ *is discrete* **then**
    $\lambda_1 \leftarrow \mu(c), \lambda_2 \leftarrow \nu(c)$
    Cost matrix:
      $\mathbf{C} := d(\cdot, \cdot) : \mathcal{C} \times \mathcal{C} \mapsto \mathbb{R}^{n_s \times n_t}$;
**else**
    $\lambda_1 = \frac{1}{n_s} \sum_i^{n_s} c_{si}, \lambda_2 = \frac{1}{n_t} \sum_j^{n_t} c_{tj}$;
    Cost matrix $\mathbf{C} : C_{ij} = d(c_{si}, c_{tj})$;
$\mathbf{K} = exp(-\mathbf{C}/\epsilon)$ // Sinkhorn param $\epsilon$
**while** *not converge* **do**
    $a_1 \leftarrow (\lambda_1/\mathbf{K}b_1), a_2 \leftarrow (\lambda_2/\mathbf{K}b_2)$;
    $\rho \leftarrow l \odot \prod_{m=1}^2 (\mathbf{K}^\top a_m)^{w_m}$;
    $b_1 \leftarrow (\rho/\mathbf{K}^\top a_1), b_2 \leftarrow (\rho/\mathbf{K}^\top a_2)$;
    $l \leftarrow \sqrt{l \odot (\rho/\mathbf{K}l)}$;
**Output:** $\rho$

---

Towards this, motivated by the maze navigation example shown in Figure 1, throughout this section, we focus on the collection of CMDPs which obeys the following assumption:

**Assumption 4.1 (Consistency and homogeneity)** *The initial state distribution is consistent with the associated context distribution, namely $p_0(s \,|\, c) = \mathbb{1}(s = c)$. In addition, the transition kernels and rewards of the CMDP are homogeneous with respect to the context. Specifically,*

$$P(s' \,|\, s, a, c_i) = P(s' \,|\, s, a, c_j), \quad R(s, a, c_i) = R(s, a, c_j), \qquad \forall c_i, c_j \in \mathcal{C}, s \in \mathcal{S}, a \in \mathcal{A}.$$

To continue, for $k$-th stage associated with the context distribution $\rho_k \in \Delta(\mathcal{C})$, we denote the optimal policy in $k$-th stage as $\pi_k^\star := \arg\max_\pi V^\pi(\rho_k)$. Armed with the above notations and assumptions, we are positioned to introduce our main theorem.

**Theorem 4.1** *Let $m$ be a problem-dependent positive constant. Consider any MDP under the assumption 4.1 and the situations further specified in Appendix A.1. For any two subsequent stages $k, k + 1 \in [K]$, the optimal policies $\pi_k^\star, \pi_{k+1}^\star$ obey*

$$V^{\pi_{k+1}^\star}(\rho_{k+1}) - V^{\pi_k^\star}(\rho_{k+1}) \leq m\mathcal{W}_{d^\star}(\rho_k, \rho_{k+1}). \tag{6}$$

*Here, $d^\star$ is the distance metric defined on the metric space $(\mathcal{C}, d^\star)$ which obeys ($\mathcal{U}(\mathcal{C})$ denote the uniform distribution over $\mathcal{C}$)*

$$\pi^\star := \max_\pi V^\pi(\mathcal{U}(\mathcal{C})), \qquad d^\star(\cdot, \cdot) := d^{\pi^\star}(\cdot, \cdot). \tag{7}$$

The proof can be found in Appendix A.4. In addition, the following corollary can be directly verified by Theorem 4.1.

**Corollary 4.1.1** *By constant speed geodesics (7.2 of [51]), the stages generated by GRADIENT obey*

$$V^{\pi_{k+1}^\star}(\rho_{k+1}) - V^{\pi_k^\star}(\rho_{k+1}) \leq m\Delta\alpha\mathcal{W}_{d^\star}(\mu, \nu). \tag{8}$$

**Remark 4.1** *Theorem 4.1 combined with Corollary 4.1.1 indicates that with tailored stages by GRADIENT (sufficiently small $\Delta\alpha$), it is easy to adapt the learned optimal policy $\pi_k^\star$ in the current stage to the optimal policy $\pi_{k+1}^\star$ in the next stage defined by $\rho_{k+1}$, since the performance gap is small. In particular, after we obtain the optimal policy $\pi_k^\star$ associated with the $k$-th stage in the curriculum and transfer this policy to the next stage, the performance gap between it and the optimal policy in the $(k+1)$-th stage can be controlled by the Wasserstein distance between two subsequent context distributions and therefore a fraction of the Wasserstein distance between the source and the target task distribution (which can be seen as a constant). By properly controlling the Wasserstein distance using GRADIENT, we can achieve gradual domain adaptation from the source to the target task distribution.*

# 5 Experiments

**Evaluation Metric.** To quantify the benefit of CRL, we compare the learning progress of an agent evaluated on the target task distribution [1]. More concretely, the metric we use is *time-to-threshold* [6], which shows how fast an agent can learn a policy that achieves a satisfying return on the target task if it transfers knowledge. We also compare the *asymptotic performance* of the agent. For the learner, we use the SAC [52] and PPO [53] implementations provided in the `StableBaseline3` library [54]. For the optimal tranport computation, we use POT [55].

**Environments.** The first **Maze** environment is to investigate the performance of `GRADIENT` and the effect of $\Delta\alpha$. Maze has a discrete context space and therefore the exact context distance metric is available. The next two environments are to investigate the benefits of `GRADI-ENT` when the context space is continuous, and we use $l_2$ distance as a surrogate contextual distance. In the **Point-Mass** environment, the source and the target distribution

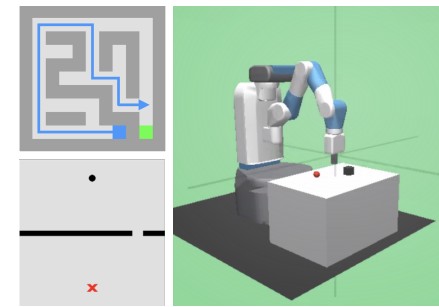

Figure 2: Environment visualizations. For the **Maze** (upper-left), the context is the initial state. For the **PointMass** (lower-left), the context is the position and width of the gap on the wall. For the **FetchPush**, the context is the goal position on a 2D surface.

are both Gaussian distributions, which are in line with the settings of existing algorithms. In the **FetchPush** environment, we want to investigate the case where the source and the target are not Gaussian distributions.

We benchmark our algorithm with six baselines, including six CRL algorithms:

1. `No-Curriculum`: sampling contexts according to the target task distribution.
2. `Domain-Randomization`: uniformly sampling the contexts from the context space.
3. `HER` [56]: implicit CRL method by relabelling trajectories.
4. `Linear-Interpolation`: sampling contexts based on a weighted average between two distributions, i.e., $(1 - \alpha)\mu + \alpha\nu$.
5. `ALP-GMM` [13]: the state-of-the-art intrinsically motivated CRL algorithm that maximizes the absolute learning progress of the agent.
6. `Goal-GAN` [12]: using GAN to generate contexts of intermediate difficulties and automatically explore the goal space.
7. `SPDL` [8] and `WB-SPDL` [57]: interpreting the curriculum generation as a variational inference problem to progressively approach the target task distribution. `WB-SPDL` extends SPDL by using Wasserstein distance instead of KL divergence.

## 5.1 Can GRADIENT Handle Discrete Contexts and What is the Effect of $\Delta\alpha$?

In the **Maze** task, we fix the layout of the maze and the goal. The state and action spaces are discrete. The action space includes movements in four directions. The context is the initial position, and therefore the context space overlaps with the state space. The observation is the flattened value representation of the maze, including the goal, the current position, and the layout.

We visualize the optimal $\pi^*$-contextual-distance metric in Figure 3a (computed using the optimal policy and normalized to make the maximum value be 1). The distance metric can be represented by a symmetrical matrix. We then generate curricula using `GRADIENT` with $\Delta\alpha = 0.2, 0.1, 0.05$. From Figure 3b, our proposed method outperforms the baselines significantly in terms of the time-to-threshold evaluated in the target distribution. With a smaller $\Delta\alpha$, `GRADIENT` learns slightly slower; nevertheless, all choices achieve good performances. In this simple environment, `Domain-Randomization` also improves the learning efficiency compared with `No-Curriculum` (which cannot learn at all). Interestingly, `Domain-Randomization` even outperforms the `Linear-Interpolation`, which demonstrates the caveat of bad intermediate task distribution.

---

[1]Code is available under https://github.com/PeideHuang/gradient.git

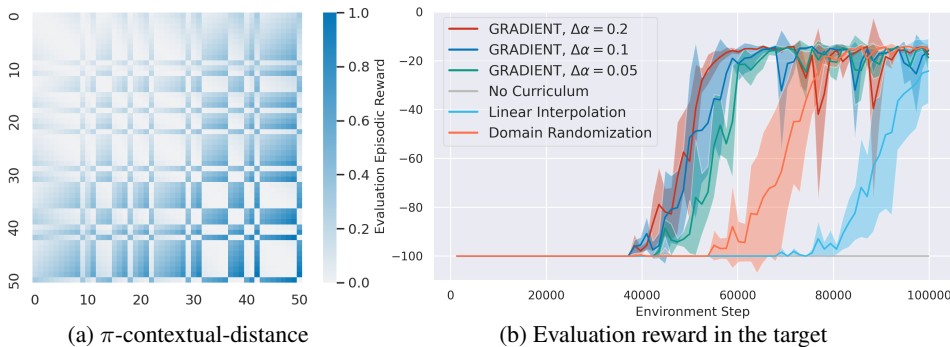

(a) $\pi$-contextual-distance

(b) Evaluation reward in the target

Figure 3: The $\pi$-contextual-distance and evaluation results in Maze. The shaded area represents the standard deviation. We use a PPO learner and show that GRADIENT converges faster than baselines.

## 5.2 How Does GRADIENT Perform When the Distributions Are Gaussian?

In PointMass [8], the agent needs to navigate a pointmass through a wall with a small gap at an off-center position to reach the goal. The context is a 2-dimension vector representing the width and position of the gap on the wall. Following the setting in the SPDL paper [8], the target distribution is an isotropic Gaussian distribution centered at $[2.5, 0.5]$ with a negligible variance $[4 \times 10^{-3}, 3.75 \times 10^{-3}]$ (which is effectively a point as shown in Figure 5a). The source distribution is an isotropic Gaussian distribution centered at $[0, 4.25]$ with a variance of $[2, 1.875]$.

From Figure 4a, we observe that both GRADIENT and SPDL significantly outperform other baselines in terms of the time-to-threshold and achieve the same asymptotic performance. It is reasonable since these two methods are capable of specifying the target distribution to guide curriculum generation. We present the curricula visualization of both methods in Figure 5a and show that the intermediate task distributions gradually move from the source to the target distribution. However, the other three baselines lack the flexibility to leverage the target distribution. This experiment highlights the importance of being able to specify the target distribution.

## 5.3 How Does GRADIENT Perform When the Distributions Are NOT Gaussian?

In FetchPush [58], the objective is to use the gripper to push the box to a goal position. The observation space is a 28-dimension vector, including information about the goal. The context is a 2-dimension vector representing the goal position on a surface. The target distribution is a uniform distribution over the circumference of a half-circle (Figure 5b). The source distribution is a uniform distribution over a square region centered at the box position, excluding the region within a certain radius of the object. We use this experiment to highlight the importance of the capability to handle arbitrary distributions rather than only the parametric Gaussian distributions. Since SPDL can only

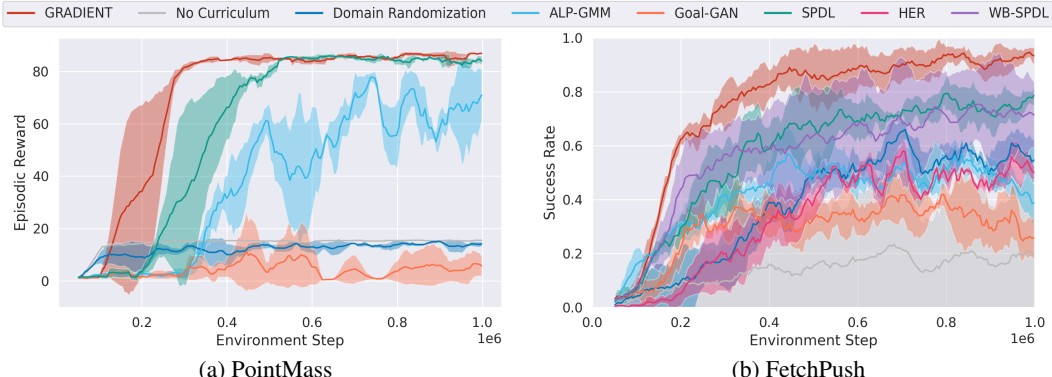

(a) PointMass

(b) FetchPush

Figure 4: Evaluation performance in the target task distribution (with a rolling average of 10) for PointMass and FetchPush. In PointMass, where both the source and target distributions are defined by a parametric Gaussian, GRADIENT converges faster than baselines. In FetchPush where the target task distribution can NOT be accurately approximated by a Gaussian, GRADIENT converges faster and has a better asymptotic performance. We use SAC as the learner for both environments.

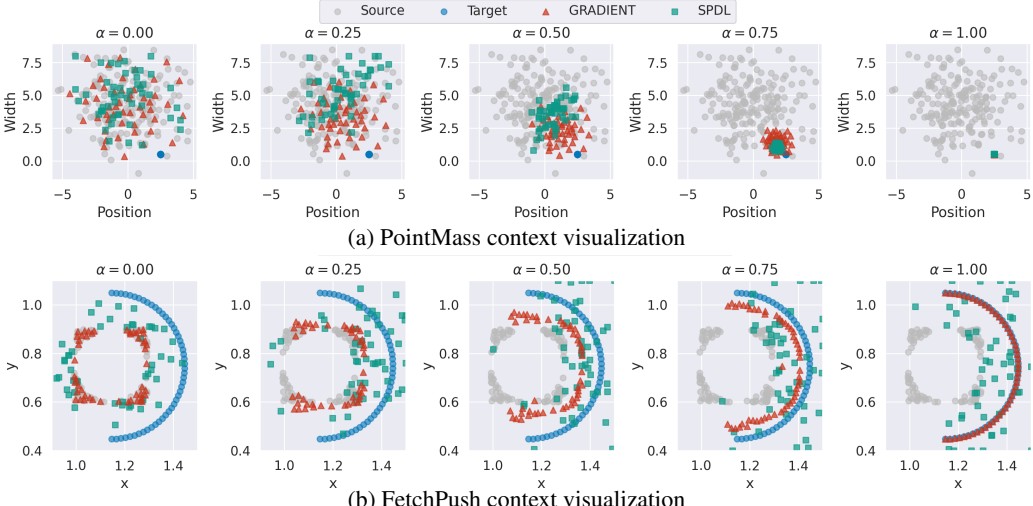

(a) PointMass context visualization

(b) FetchPush context visualization

Figure 5: Visualizations of context distributions and curricula in PointMass and FetchPush. We present GRADIENT's curricula corresponding to $\alpha = 0.00, 0.25, 0.50, 0.75, 1.00$ together with SPDL's contexts from 5 representative stages. For PointMass, the contexts of SPDL are taken from environment steps at 10k, 50k, 100k, 200k, and 300k. For FetchPush, the contexts of SPDL are taken from environment steps at 5k, 50k, 100k, 250k, and 500k.

deal with parametric Gaussian distribution, we first fit the target and the source distribution with two Gaussian distributions and feed them into the baseline algorithms.

From Figure 4b, GRADIENT outperforms all the baseline methods in terms of the time-to-threshold in the target distribution. For ALP-GMM and Goal-GAN, since these two methods do not consider the target distribution, they achieve lower performance than the target-aware methods such as SPDL and GRADIENT. We further compare these two target-aware methods and find that GRADIENT outperforms the SPDL in terms of learning efficiency and asymptotic performance. The reason can be explained by visualizing the context distribution in Figure 5b. SPDL indeed shows the behavior of moving the context distribution towards the target. However, since the target distribution cannot be fit with a Gaussian well, many sampled contexts are in fact far away from the target distribution, which may not improve the learning efficiency in the target domain. Although Domain-Randomization learns slightly lower than the CRL methods, it achieves even higher asymptotic performance than ALP-GMM and Goal-GAN. The potential reason could be that it does not bias towards a certain area while it is possible for ALP-GMM and Goal-GAN.

## 5.4 Beyond Euclidean Distance Metric: Using Distance Embedding for Continuous Spaces

In practice, it is not always appropriate to use the Euclidean distance as the surrogate contextual distance. Though there are many existing OT algorithms based on Euclidean distance metric, solving for free-support Wasserstein barycenters on non-Euclidean space is non-trivial [59, 60]. One solution to bypass this issue is to embed contexts to a latent space such that the Euclidean interpolation in the latent space approximates the interpolation in the original non-Euclidean space after reconstruction.

For the fixed-support setting, Deep Wasserstein Embedding (DWE) proposed in [61] uses Siamese networks [62] to learn a mapping function such that the Euclidean distance in the latent space approximates the Wasserstein distance in the original space. Motivated by this idea, we instead develop an embedding mechanism for free supports, i.e., continuous contexts. We leverage an encoder to embed contexts to latent space in which the Euclidean distance mimics the original distance metric. We train the encoder and decoder based on a reconstruction loss and a distance embedding loss. We include the details of distance embedding in Algorithm 5 and a modified version of GRADIENT in Algorithm 4 in Appendix C.2.

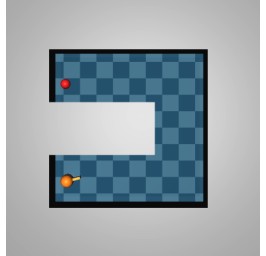

Figure 6: U-Maze environment. The orange and red balls represent the agent and the goal, respectively. The context is the goal position.

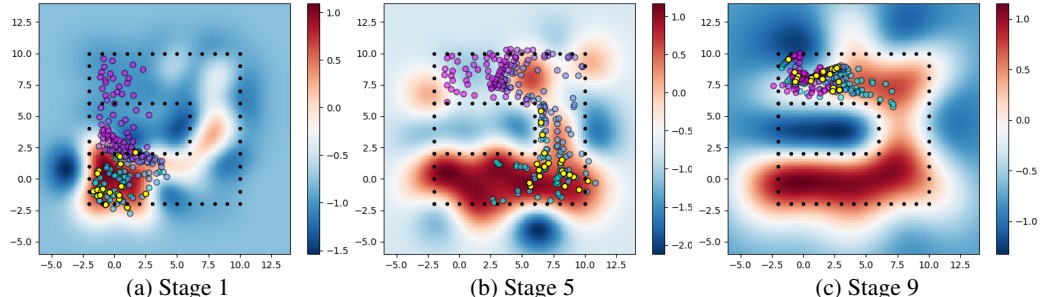

|                  |                  |                  |
|:----------------:|:----------------:|:----------------:|
| (a) Stage 1      | (b) Stage 5      | (c) Stage 9      |

Figure 7: U-Maze interpolation results. The original source and target distributions are two Gaussian distributions centered at $[0, 0]$ and $[0, 8]$. The color of the heat map represents the estimate of $J^\pi(c)$. The colored circles represent the Wasserstein interpolations. More specifically, the circles with cyan to purple color represent the potential interpolation results from the current source to target with $\alpha = 0.0, 0.1, 0.2, \ldots, 1.0$. The yellow circles highlight the selected barycenter corresponding to the current stage.

We show an example of using Algorithm 4. Here we consider a classical U-shaped maze with continuous spaces in Figure 6. We assume that the source and target distributions are two Gaussian distributions at the two ends of the maze. Due to the existence of the obstacle in the middle, it is not appropriate to use the $l_2$ distance as the contextual distance. In this case, we use $d^\pi(c_1, c_2) := |J^\pi(c_1) - J^\pi(c_2)|$ as a surrogate metric for the $\pi$-contextual-distance, where $J^\pi(c)$ represents the expected episodic reward of context $c$ under the policy $\pi$.

The interpolation results are shown in Figure 7. At the first few stages, since the agent does not have a good policy or a good estimate of the episodic reward, the potential geodesic interpolation tries to go through the obstacle in the middle. Fortunately, since $\alpha$ is small at first, the selected intermediate barycenters (yellow circles) are relatively close to the source distribution, so GRADIENT will not generate unreasonable tasks. As the learning progressing, the estimates become better and better and therefore produce better interpolation results.

## 6 Conclusion and Limitation

In this work, we propose a novel curriculum reinforcement learning framework as an optimal transport problem. We also develop a distance metric to measure the difference between tasks. We show that our proposed method is able to achieve smooth transfer between subsequent stages in the generated curriculum. Finally, the empirical results highlight the three features of our methods: considering metric information, being aware of the target, and being able to handle nonparametric distributions. For future work, it could be interesting to investigate how to use the *intrinsic curiosity* driven type of method as in [22, 13, 23] to automatically generate the source distribution. In addition, wider choices of the context distance can be explored to provide more insights into the OT-based CRL methods. From a theoretical perspective, OT also provides powerful mathematical tools for future researches to investigate rigorously why CRL methods can help reduce the sample complexity.

There are two main limitations of our proposed GRADIENT. First, due to the limitation of OT computation complexity, we only examine the settings that are either discrete or low-dimensional. Such experiment design choice is common in the related works and baselines [13, 12, 8]. It comes from both the fact of many tasks can be represented by a low-dimensional vector, such as goal positioned or physical parameters of robots. We believe how to deal with high-dimensional contexts, such as images, is a promising future direction. Second, our analysis is conducted based on a restrictive subset of contextual MDPs (Assumption 4.1)where the context controls only the initial state distribution. It would be beneficial to relax this assumption by imposing assumptions on the specific manner in which the context controls the dynamics or reward.

## Acknowledgments

We gratefully acknowledge support from the National Science Foundation under grant CAREER CNS-2047454. Fei Fang was supported in part by NSF grant IIS-2046640 (CAREER).

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
