# A Proof of Theorem 4.1

In this section, we shall provide the proof for Theorem 4.1. The rest of this section is organized as follows. Appendix A.1 provides additional useful notations and definitions including but not limited to CMDPs, value function, and distance metrics. Appendix A.2 introduces further assumptions, and A.3 introduces the preliminary of optimal transport. Then, we provide the proof pipeline of Theorem 4.1 in Appendix A.4 and postpone the auxiliary proof to Appendix A.5. Finally, we summarize all the useful notations in Table 1.

## A.1 Additional notation and definitions used in the proof

Before starting, let's introduce some additional notations useful throughout the theoretical analysis. For any discrete set $X$, we will denote the set of probability measures on $X$ by $\Delta(X)$. For any vector $x \in \mathbb{R}^{SA}$ (resp. $x \in \mathbb{R}^S$ or $x \in \mathbb{R}^C$) that constitutes certain values for each state-action pair (resp. state or context), we use $x(s,a)$ (resp. $x(s)$ or $x(c)$) to denote the entry associated with the $(s,a)$ pair (resp state $s$ or context $c$). We denote $\mathsf{supp}(\rho)$ as the support of any distribution $\rho$.

**Target CMDPs.** Throughout the proof, we shall focus on the set of CMDPs introduced in Assumption 4.1. Moreover, without loss of generality, let the state space $\mathcal{S} = \{1, 2, \cdots, S\}$ (resp. action space $\mathcal{A} = \{1, 2, \cdots, A\}$) to be with finite cardinality $S$ (resp. $A$). Without loss of generality, let the immediate reward $R(s, a, s) \in [0, 1-\gamma]$ for all $(s, a) \in \mathcal{S} \times \mathcal{A}$.

First, we introduce the following *occupancy distributions* associated with policy $\pi$ and any CMDP with the initial state distribution $c \sim \rho$:

$$o^\pi(s; \rho) := (1-\gamma) \sum_{t=0}^\infty \gamma^t \mathbb{P}(s_t = s \mid c \sim \rho; \pi), \tag{9a}$$

$$o^\pi(s, a; \rho) := (1-\gamma) \sum_{t=0}^\infty \gamma^t \mathbb{P}(s_t = s, a_t = a \mid c \sim \rho; \pi) = o^\pi(s; \rho)\pi(a \mid s). \tag{9b}$$

With this in mind, we introduce two important definitions in the proof,: the covering state-action (resp. state) space by executing $\pi$ with the initial state distribution $c \sim \rho$ defined as

$$\mathcal{C}(\rho, \pi) = \{(s, a) : o^\pi(s, a; \rho) > 0\}, \qquad \mathcal{C}_s(\rho, \pi) = \{s : o^\pi(s; \rho) > 0\}. \tag{10}$$

The covering space represent the area that is possible to be visited by the policy $\pi$ given the initial state distribution.

**Value function.** Different from the value function defined by context distribution $\rho$, with abuse of notation, we denote the value function $V_{\mathsf{s}}$ associated with a state $s$, a policy $\pi$ and the target CMDPs as

$$V_{\mathsf{s}}^\pi(s) = \mathbb{E}\left[\sum_{t=0}^\infty \gamma^t R(s_t, a_t, c') \mid s_0 = s, c = s; \pi\right]. \tag{11}$$

For convenience, for any $s \in \mathcal{S}$, we define $\rho_s(c) := \mathbb{1}(c = s)$. Recalling that the target MDPs are defined by $p_0(s \mid c) = \mathbb{1}(s = c)$, it is easily verified that for any policy $\pi$, the value function satisfies the following properties:

$$\forall s \in \mathcal{S}: \qquad\qquad 0 \le V_{\mathsf{s}}^\pi(s) = V^\pi(\rho_s) \le 1, \tag{12a}$$

$$\forall \rho \in \Delta(\mathcal{C}) \in \Delta(\mathcal{S}): \qquad\qquad V_{\mathsf{s}}^\pi(\rho) = V^\pi(\rho). \tag{12b}$$

**Distance metrics.** Furthermore, for the specified target CMDPs, plugging in $p_0(s \mid c) = \mathbb{1}(s = c)$, for any two contexts $c_i, c_j \in \mathcal{C}$, the $\pi$**-contextual-distance** metric defined in context space $\mathcal{C}$ obeys

$$d^\pi(c_i, c_j) = \mathbb{E}_{s_i \sim p_0(\cdot \mid c_i), s_j \sim p_0(\cdot \mid c_j)}\left[d_{c_i, c_j}^\pi(s_i, s_j)\right] = d_{s_i, s_j}^\pi(s_i, s_j), \tag{13}$$

where $s_i$ (resp. $s_j$) is the corresponding initial state when the context are $c_i$ (resp. $c_j$). With this observation, we highlight that for the target CMDPs, the **contextual $\pi$-bisimulation** metric degrades to **$\pi$-bisimulation** $d_{\mathsf{s}}^\pi(\cdot, \cdot)$ [48] such that:

$$\forall s_i, s_j \in \mathcal{S}: \qquad d_{\mathsf{s}}^\pi(s_i, s_j) := d_{s_i, s_j}^\pi(s_i, s_j) = d^\pi(s_i, s_j). \tag{14}$$

The above result directly leads to that for any $\rho, \rho' \in \Delta(\mathcal{C}) \subseteq \Delta(\mathcal{S})$,

$$\mathcal{W}_{d^\pi}(\rho, \rho') = \mathcal{W}_{d_{\mathsf{s}}^\pi}(\rho, \rho'). \tag{15}$$

## A.2 Additional Assumptions in the proof

Besides the key properties of the target CMDPs introduced in Assumption 4.1, we shall introduce some auxiliary assumptions for convenience as follows:

- **Bounded environment.** For the optimal policy $\pi^\star$ and any state $s \in \mathcal{S}$, without loss of generality, the minimum and maximum $\pi$-**bisimulation** distance between $s$ and other states satisfies

$$\min_{s' \in \mathcal{S}} d_s^\star(s, s') \, \mathbb{1}(d_s^\star(s, s') > 0) \geq 1, \qquad \frac{D_{\mathsf{max}}}{2} \leq \max_{s' \in \mathcal{S}} d_s^\star(s, s') \leq D_{\mathsf{max}}, \qquad (16)$$

for some universal positive constant $D_{\mathsf{max}}$.

- **Deterministic environment.** Without loss of generality, we suppose the transition kernel is deterministic. Moreover, at any state $s \in \mathsf{supp}(\rho_{k+1}) \setminus \mathsf{supp}(\rho_k)$, there are a fixed portion of actions lead to the decreasing of the distance $\min_{s' \in \rho_k} d_s^\star(s', s)$, i.e., for any transition sample $(s_t, a_t, s_{t+1})$,

$$\min_{s' \in \rho_k} d_s^\star(s', s_{t+1}) = \begin{cases} \min_{s' \in \rho_k} d_s^\star(s', s_t) - 1 & \text{with probability p} \\ \min_{s' \in \rho_k} d_s^\star(s', s_t) + 1 & \text{otherwise.} \end{cases} \qquad (17)$$

- **Closeness of the subsequent stages.** Starting from any state $s_0 = t \in \mathsf{supp}(\rho_{k+1}) \setminus \mathsf{supp}(\rho_k)$, if the sample trajectory shall enter $\mathcal{C}_s(\rho_k, \pi_k^\star)$ before arriving at somewhere $s$ obeying $d_s^\star(s, t) = \max_{s \in \mathcal{S}} d_s^\star(s, t)$. Then we suppose that the hitting point $s$ is not far from the initial distribution $\rho_k$ of the previous stage $k$, namely

$$\mathbb{E}_s[d_s^\star(s, t)] \leq C_1 \min_{s' \in \rho_k} d_s^\star(s', t), \qquad (18)$$

for some universal constant $C_1 > 0$.

We would like to note that these additional assumptions are useful for a concise proof to show the main intuition and idea of our proposed algorithm **GRADIENT**. These are some general assumptions without specific limitations. It is interesting to extend our main Theorem 4.1 to more general cases which shall be the further work.

## A.3 Preliminaries: optimal transport and random walk

In this subsection, we introduce several key properties/facts of optimal transport, value function, and random walk, which play a crucial role in the proof of Theorem 4.1. The proofs for this subsection are deferred to Appendix A.5.

**Preliminaries of optimal transport.** We first introduce an important fact of the optimization problem in optimal transport. Suppose there exists two discrete sets $\mathcal{X}, \mathcal{Y}$ (possibly different). As it is well-known, for any two distributions $\mu \in \Delta(\mathcal{X}), \nu \in \Delta(\mathcal{Y})$, 1-Wasserstein distance with any metric $d$ between $\mu$ and $\nu$ can be expressed as the optimal solution of the following linear programming (Kantorovich duality) problem:

$$\max_{u,v} \sum_{x \in \mathcal{X}} u(x)\mu(x) - \sum_{y \in \mathcal{Y}} v(y)\nu(y),$$
$$\text{subject to} \quad \forall x \in \mathcal{X}, y \in \mathcal{Y}: \ u(x) - v(y) \leq d(x, y),$$
$$0 \leq u \leq 1, 0 \leq v \leq 1. \qquad (19)$$

Note that, when $\mathcal{X} = \mathcal{Y} = \mathcal{S}$, the resulting claim in (19) is equivalent to [48, 63, 64].

**Controlling performance gap.** Inspired by the above problem formulation, the difference of the value function conditioned on two different states/state distribution can be controlled in the following lemma:

**Lemma A.1** *For any $s, t \in \mathcal{S}$ and any policy $\pi$, we have*

$$|V_s^\pi(s) - V_s^\pi(t)| \leq d_s^\pi(s, t). \qquad (20)$$

*In addition, for any two context distribution $\rho_1, \rho_2 \in \Delta(\mathcal{C}) \in \Delta(\mathcal{S})$, the performance gap associated with any policy $\pi$ obeys*

$$|V_s^\pi(\rho_1) - V_s^\pi(\rho_2)| \leq \mathcal{W}_{d^\pi}(\rho_1, \rho_2). \qquad (21)$$

This fact connects the performance gap in RL and the Wasserstein distance between different state distribution/contextual distribution, taking advantage of the Kantorovich duality form of the optimal transport problem.

**Definition and properties of random walk.** Finally, we describe the following useful lemma which is essential in proving the main part of Theorem 4.1 when the starting state is not in the covering area of the optimal policy of the previous training stage

**Lemma A.2** *The process* $\{S_n : n \geq 1\}$ *is called a random walk if* $\{X_i : i \geq 1\}$ *are iid Bernoulli distribution with* $p < 1$ *and* $S_n = \sum_{i=1}^{n} X_i$. *Starting from 0, the expectation of the* **stopping time** $N$ *of hitting any* $d > 0$ *or* $-D_{\mathsf{max}}$ *and the probability* $p_\alpha$ *of hitting* $d > 0$ *obeys:*

$$\mathbb{E}[N] \leq \max\left\{D_{\mathsf{max}}d\,,\ \sqrt{\frac{D_{\mathsf{max}}}{(1-2p)\wedge 1}}d\right\}, \quad p_\alpha \geq 1 - \frac{2d\sqrt{D_{\mathsf{max}}}}{d + D_{\mathsf{max}}}. \tag{22}$$

### A.4 Proof of Theorem 4.1

With above preliminaries in hand, we are ready to embark on the proof for Theorem 4.1, which is divided into multiple steps as follows.

**Step 1: decomposing the performance gap of interest.** To begin with, we decompose the term of interest as follows:

$$
\begin{aligned}
V^{\pi_{k+1}^\star}(\rho_{k+1}) - V^{\pi_k^\star}(\rho_{k+1}) &\overset{(i)}{=} V_\mathsf{s}^{\pi_{k+1}^\star}(\rho_{k+1}) - V_\mathsf{s}^{\pi_k^\star}(\rho_{k+1}) \\
&\overset{(ii)}{=} V_\mathsf{s}^{\pi^\star}(\rho_{k+1}) - V_\mathsf{s}^{\pi_k^\star}(\rho_{k+1}) \\
&= V_\mathsf{s}^{\pi^\star}(\rho_{k+1}) - V_\mathsf{s}^{\pi^\star}(\rho_k) + V_\mathsf{s}^{\pi^\star}(\rho_k) - V_\mathsf{s}^{\pi_k^\star}(\rho_{k+1}) \\
&\overset{(iii)}{\leq} \mathcal{W}_{d^\star}(\rho_{k+1}, \rho_k) + V_\mathsf{s}^{\pi_k^\star}(\rho_k) - V_\mathsf{s}^{\pi_k^\star}(\rho_{k+1}),
\end{aligned}
\tag{23}
$$

where (i) holds by the equivalence verified in (12b), (ii) follows from the value function of the optimal policy $\pi^\star$ (covering the entire state-action space) is the same as that of $\pi_{k+1}^\star$ in the region $\mathcal{C}(\rho_{k+1}, \pi_{k+1}^\star)$, and (iii) comes from applying Lemma A.1 and the fact that $\pi^\star$ achieves the same value function as $\pi_k^\star$ in the region $\mathcal{C}(\rho_k, \pi_k^\star)$.

**Step 2: controlling the second term in** (23) **in two cases.** To proceed, it is observed that

$$\forall(s,t) \in \mathcal{C}_s(\rho_k, \pi_k^\star) \times \rho_{k+1}: \qquad V_\mathsf{s}^{\pi_k^\star}(s) - V_\mathsf{s}^{\pi_k^\star}(t) \leq d_\mathsf{s}^\star(s,t) \tag{24}$$

can directly leads to that $(u,v) = \left(V_\mathsf{s}^{\pi_k^\star}, V_\mathsf{s}^{\pi_k^\star}\right)$ is a feasible solution to the Wasserstein distance $\mathcal{W}_{d^\star}(\rho_k, \rho_{k+1})$ dual formulation in (19), which yields

$$
\begin{aligned}
V_\mathsf{s}^{\pi_k^\star}(\rho_k) - V_\mathsf{s}^{\pi_k^\star}(\rho_{k+1}) &= \sum_{s\in\mathcal{C}_s(\rho_k,\pi_k^\star)} V_\mathsf{s}^{\pi_k^\star}(s)\rho_k(s) - \sum_{s\in\mathcal{C}_s(\rho_{k+1},\pi_k^\star)} V_\mathsf{s}^{\pi_k^\star}(s)\rho_{k+1}(s) \\
&\leq \mathcal{W}_{d_\mathsf{s}^\star}(\rho_k, \rho_{k+1}).
\end{aligned}
\tag{25}
$$

As a result, we turn to show (24) instead of controlling $V_\mathsf{s}^{\pi_k^\star}(\rho_k) - V_\mathsf{s}^{\pi_k^\star}(\rho_{k+1})$. Towards this, we start from considering $V_\mathsf{s}^{\pi_k^\star}(s) - V_\mathsf{s}^{\pi_k^\star}(t)$ for any $s \in \rho_k$ and $t \in \rho_{k+1}$ in two different cases: (i) $t \in \mathcal{C}_\mathsf{s}(\rho_k, \pi_k^\star)$; (ii) otherwise.

In the first case when $t \in \mathcal{C}_\mathsf{s}(\rho_k, \pi_k^\star)$, since $\pi_k^\star$ is the same as the optimal policy $\pi^\star$ in $\mathcal{C}_\mathsf{s}(\rho_k, \pi_k^\star)$, invoking Lemma A.1 directly yields

$$V_\mathsf{s}^{\pi_k^\star}(s) - V_\mathsf{s}^{\pi_k^\star}(t) = V_\mathsf{s}^{\pi^\star}(s) - V_\mathsf{s}^{\pi^\star}(t) \leq d_\mathsf{s}^\star(s,t). \tag{26}$$

**Step 3: focusing on case (ii).** Then, we shall focus on the other case when

$$t \in \mathsf{supp}(\rho_{k+1}) \setminus \mathcal{C}_\mathsf{s}(\rho_k, \pi_k^\star). \tag{27}$$

Invoking Lemma A.2, without loss of generality, since in the stage $k$, we can't visit the region outside of $\mathcal{C}_\mathsf{s}(\rho_k, \pi_k^\star)$, we can define $\pi_k^\star(\cdot \mid s)$ to be uniformly random in the unseen region, i.e.,

$$\forall a \in \mathcal{A}, s \in \mathsf{supp}(\rho_{k+1}) \setminus \mathcal{C}_\mathsf{s}(\rho_k, \pi_k^\star) : \qquad \pi_k^\star(a \mid s) = \frac{1}{A}. \tag{28}$$

Then we define $p_\mathsf{g}$ as the probability of the benign events $B$ when we can hit some point $s_\mathsf{b}$ at the boundary of the region $\mathcal{C}_\mathsf{s}(\rho_k, \pi_k^\star)$ visited by the previous stage $k$ before arriving at some limit point $s_\mathsf{limit} := \arg\max_{s' \in \mathcal{S}} d_\mathsf{s}^\star(t, s')$. Invoking the assumption of the target CMDPs in (17), combind with the policy defined in (28), we observe that applying Lemma A.2 leads to

$$p_\mathsf{g} \geq 1 - \frac{2d_\mathsf{s}^\star(t, s_\mathsf{b})\sqrt{D_\mathsf{max}}}{d_\mathsf{s}^\star(t, s_\mathsf{b}) + D_\mathsf{max}} \geq 1 - \frac{2d_\mathsf{s}^\star(t, s_\mathsf{b})}{\sqrt{D_\mathsf{max}}} \geq 1 - \frac{2C_1 d_\mathsf{s}^\star(t, s)}{\sqrt{D_\mathsf{max}}}, \tag{29}$$

where the last inequality holds by the assumption in (16).

To continue, we express the value function at state $t$ as

$$V_\mathsf{s}^{\pi_k^\star}(t) \geq p_\mathsf{g} \left( \mathbb{E}_B[\sum_{t=0}^{N} 0 \cdot \gamma^t] + \gamma^N V_\mathsf{s}^{\pi_k^\star}(s_\mathsf{b}) \right) - (1 - p_\mathsf{g}) = \gamma^N p_\mathsf{g} V_\mathsf{s}^{\pi_k^\star}(s_\mathsf{b}) - (1 - p_\mathsf{g}). \tag{30}$$

Similarly, we have

$$V_\mathsf{s}^{\pi_k^\star}(s) \leq (1 - p_\mathsf{g}) + p_\mathsf{g}\mathbb{E}[\sum_{t=0}^{N} (1 - \gamma)\gamma^t + \gamma^N V_\mathsf{s}^{\pi_k^\star}(s_N)]. \tag{31}$$

We shall control the performance gap in two cases separately:

- When $V_\mathsf{s}^{\pi_k^\star}(s_\mathsf{b}) \geq V_\mathsf{s}^{\pi_k^\star}(s_N)$. we have

$$V_\mathsf{s}^{\pi_k^\star}(s) - V_\mathsf{s}^{\pi_k^\star}(t) \leq V_\mathsf{s}^{\pi_k^\star}(s) - \gamma^N p_\mathsf{g} V_\mathsf{s}^{\pi_k^\star}(s') + (1 - p_\mathsf{g})$$

$$\leq p_\mathsf{g}\mathbb{E}[\sum_{t=0}^{N} (1 - \gamma)\gamma^t + \gamma^N V_\mathsf{s}^{\pi_k^\star}(s_N)] - \gamma^N p_\mathsf{g} V_\mathsf{s}^{\pi_k^\star}(s_\mathsf{b}) + 2(1 - p_\mathsf{g})$$

$$\leq p_\mathsf{g}\mathbb{E}[\sum_{t=0}^{N} (1 - \gamma)\gamma^t] + 2(1 - p_\mathsf{g}). \tag{32}$$

- When $V_\mathsf{s}^{\pi_k^\star}(s_\mathsf{b}) < V_\mathsf{s}^{\pi_k^\star}(s_N)$, we have

$$V_\mathsf{s}^{\pi_k^\star}(s) - V_\mathsf{s}^{\pi_k^\star}(t) \leq p_\mathsf{g}\mathbb{E}[\sum_{t=0}^{N} (1 - \gamma)\gamma^t + \gamma^N V_\mathsf{s}^{\pi_k^\star}(s_N)] - \gamma^N p_\mathsf{g} V_\mathsf{s}^{\pi_k^\star}(s_\mathsf{b}) + 2(1 - p_\mathsf{g})$$

$$\leq p_\mathsf{g}\mathbb{E}[\sum_{t=0}^{N} (1 - \gamma)\gamma^t + \gamma^N (N + V_\mathsf{s}^{\pi_k^\star}(s))] - \gamma^N p_\mathsf{g} V_\mathsf{s}^{\pi_k^\star}(s_\mathsf{b}) + 2(1 - p_\mathsf{g})$$

$$\leq p_\mathsf{g}\mathbb{E}\left[\sum_{t=0}^{N} (1 - \gamma)\gamma^t + \gamma^N (N + V_\mathsf{s}^{\pi_k^\star}(s_\mathsf{b}) + (C_1 + 1)d^\star(s, t))\right]$$

$$- \gamma^N p_\mathsf{g} V_\mathsf{s}^{\pi_k^\star}(s_\mathsf{b}) + 2(1 - p_\mathsf{g})$$

$$\leq p_\mathsf{g}\left[\sum_{t=0}^{N} (1 - \gamma)\gamma^t + \gamma^N (N + (C_1 + 1)d^\star(s, t))\right] + 2(1 - p_\mathsf{g}). \tag{33}$$

Summing up the above two cases yields, for all $s \in \mathcal{C}(\rho_k, \pi_k^\star), t \in \mathsf{supp}(\rho_{k+1}) \setminus \mathcal{C}(\rho_k, \pi_k^\star)$,

$$
V_{\mathsf{s}}^{\pi_k^\star}(s) - V_{\mathsf{s}}^{\pi_k^\star}(t)
$$

$$
\leq p_{\mathsf{g}} \left[ \sum_{t=0}^{N}(1-\gamma)\gamma^t + \gamma^N \left( N + (C_1+1)d^\star(s,t) \right) \right] + 2(1-p_{\mathsf{g}})
$$

$$
\leq p_{\mathsf{g}} \mathbb{E}_B[N] + p_{\mathsf{g}}(C_1+1)d^\star(s,t) + 2(1-p_{\mathsf{g}})
$$

$$
\overset{(i)}{\leq} \max\left\{ D_{\mathsf{max}}d^\star(s,t), \sqrt{\frac{D_{\mathsf{max}}}{(1-2p)\wedge 1}} d^\star(s,t) \right\} + (C_1+1)d^\star(s,t) + \frac{4d^\star(s,t)}{\sqrt{D_{\mathsf{max}}}}
$$

$$
\leq \left( \max\left\{ D_{\mathsf{max}}, \sqrt{\frac{D_{\mathsf{max}}}{(1-2p)\wedge 1}} \right\} + (C_1+1) + \frac{4}{\sqrt{D_{\mathsf{max}}}} \right) d^\star(s,t)
$$

$$
\leq c_{\mathsf{model}}d^\star(s,t) \tag{34}
$$

where (i) follows from Lemma A.2, and the last inequality holds by choosing

$$
c_{\mathsf{model}} := \left( \max\left\{ D_{\mathsf{max}}, \sqrt{\frac{D_{\mathsf{max}}}{(1-2p)\wedge 1}} \right\} + (C_1+1) + \frac{4}{\sqrt{D_{\mathsf{max}}}} \right)
$$

as a large enough problem-dependent parameter.

**Step 4: summing up the results.** Combining the results (26) and (34) in two cases, we directly arrive at, for all $s \in \mathcal{C}(\rho_k, \pi_k^\star), t \in \mathsf{supp}(\rho_{k+1})$, the following fact is satisfied

$$
V_{\mathsf{s}}^{\pi_k^\star}(s) - V_{\mathsf{s}}^{\pi_k^\star}(t) \leq c_{\mathsf{model}}d^\star(s,t),
$$

$$
\rightarrow \frac{1}{c_{\mathsf{model}}}V_{\mathsf{s}}^{\pi_k^\star}(s) - \frac{1}{c_{\mathsf{model}}}V_{\mathsf{s}}^{\pi_k^\star}(t) \leq d^\star(s,t). \tag{35}
$$

We observe that $(u,v) = \left( \frac{1}{c_{\mathsf{model}}}V_{\mathsf{s}}^{\pi_k^\star}, \frac{1}{c_{\mathsf{model}}}V_{\mathsf{s}}^{\pi_k^\star} \right)$ is a feasible solution to the Wasserstein distance $\mathcal{W}_{d_{\mathsf{s}}^\star}(\rho_k, \rho_{k+1})$ dual formulation in (19), we achieve

$$
V_{\mathsf{s}}^{\pi_k^\star}(\rho_k) - V_{\mathsf{s}}^{\pi_k^\star}(\rho_{k+1}) \leq c_{\mathsf{model}}\mathcal{W}_{d_{\mathsf{s}}^\star}(\rho_k, \rho_{k+1}). \tag{36}
$$

Finally, plugging (36) into (23) complete the proof by

$$
V^{\pi_{k+1}^\star}(\rho_{k+1}) - V^{\pi_k^\star}(\rho_{k+1}) \leq (1+c_{\mathsf{model}})\mathcal{W}_{d^\star}(\rho_k, \rho_{k+1}). \tag{37}
$$

### A.5 Proof of auxiliary lemmas

**Proof of Lemma A.1.** As it is well known that $V_{\mathsf{s}}^\pi(s)$ is the fixed point of the Bellman operator $\mathcal{T}(V,\pi) := \mathbb{E}_{a\sim\pi(\cdot\,|\,s)}[R(s,a,s) + \gamma \sum_{s'} P(s'\,|\,s,a,s)V]$. In addition, $d_{\mathsf{s}}^\pi$ is the fixed point of the operator [48]

$$
\mathcal{F}(d)(s,t) := \left| \mathbb{E}_{a\sim\pi(\cdot\,|\,s)}[R(s,a,s) - \mathbb{E}_{a\sim\pi(\cdot\,|\,t)}[R(t,a,t)] \right|
$$

$$
+ \gamma\mathcal{W}_d\left( \mathbb{E}_{a\sim\pi(\cdot\,|\,t)}[P(\cdot\,|\,s,a,s)], \mathbb{E}_{a\sim\pi(\cdot\,|\,t)}[P(\cdot\,|\,t,a)] \right). \tag{38}
$$

Armed with above facts, initializing $V_{\mathsf{s},0}^\pi = 0$ and $d_{\mathsf{s},0}^\pi = 0$, the update rules of $V_{\mathsf{s},n+1}^\pi$ and $d_{\mathsf{s},n+1}^\pi$ at the $(n+1)$-th iteration are defined as

$$
V_{\mathsf{s},n+1}^\pi = \mathcal{T}(V_{\mathsf{s},n}^\pi, \pi),
$$

$$
d_{\mathsf{s},n+1}^\pi = \mathcal{F}(d_{\mathsf{s},n}^\pi)(s,t). \tag{39}
$$

With this in mind, for any $s, t \in \mathcal{S}$, we arrive at,

$$\left| V_{\mathsf{s},n+1}^{\pi}(s) - V_{\mathsf{s},n+1}^{\pi}(t) \right|$$

$$= \left| \mathbb{E}_{a \sim \pi(\cdot \,|\, s)} \left[ R(s, a, s) + \gamma \sum_{s'} P(s' \,|\, s, a, s) V_{\mathsf{s},n}^{\pi} \right] \right.$$

$$\left. - \mathbb{E}_{a \sim \pi(\cdot \,|\, t)} \left[ R(t, a, t) + \gamma \sum_{s'} P(s' \,|\, t, a, t) V_{\mathsf{s},n}^{\pi} \right] \right|$$

$$\leq \left| \mathbb{E}_{a \sim \pi(\cdot \,|\, s)} R(s, a, s) - \mathbb{E}_{a \sim \pi(\cdot \,|\, t)} R(t, a, t) \right| + \gamma \left| \sum_{s'} P(s' \,|\, s, a, s) V_{\mathsf{s},n}^{\pi} - \sum_{s'} P(s' \,|\, t, a, t) V_{\mathsf{s},n}^{\pi} \right|$$

$$\overset{(i)}{\leq} \left| \mathbb{E}_{a \sim \pi(\cdot \,|\, s)} R(s, a, s) - \mathbb{E}_{a \sim \pi(\cdot \,|\, t)} R(t, a, t) \right|$$

$$+ \gamma \mathcal{W}_{d_{\mathsf{s},n}^{\pi}} \left( \mathbb{E}_{a \sim \pi(\cdot \,|\, s)} [P(s' \,|\, s, a, s)], \mathbb{E}_{a \sim \pi(\cdot \,|\, t)} [P(s' \,|\, t, a, t)] \right)$$

$$= \mathcal{F}(d_{\mathsf{s},n}^{\pi})(s, t) = d_{\mathsf{s},n+1}^{\pi}, \tag{40}$$

where (i) holds by $(u, v) = \left( V_{\mathsf{s},n}^{\pi}, V_{\mathsf{s},n}^{\pi} \right)$ is a feasible solution to the Wasserstein distance $\mathcal{W}_{d_{\mathsf{s},n}^{\pi}} \left( \mathbb{E}_{a \sim \pi(\cdot \,|\, s)} [P(s' \,|\, s, a, s)], \mathbb{E}_{a \sim \pi(\cdot \,|\, t)} [P(s' \,|\, t, a, t)] \right)$ dual formulation in (19).

As a result, we have

$$\forall n = 1, 2, \cdots: \qquad \left| V_{\mathsf{s},n+1}^{\pi}(s) - V_{\mathsf{s},n+1}^{\pi}(t) \right| \leq d_{\mathsf{s},n+1}^{\pi}, \tag{41}$$

which directly yields

$$\forall s, t \in \mathcal{S}: \qquad |V_{\mathsf{s}}^{\pi}(s) - V_{\mathsf{s}}^{\pi}(t)| \leq d_{\mathsf{s}}^{\pi}(s, t). \tag{42}$$

The above fact indicates that for any two distribution $\rho, \rho' \in \Delta(\mathcal{S})$,

$$V_{\mathsf{s}}^{\pi}(\rho_k) - V_{\mathsf{s}}^{\pi}(\rho_{k+1}) = \sum_{s \in \mathcal{S}} V_{\mathsf{s}}^{\pi}(s) \rho(s) - \sum_{s \in \mathcal{S}} V_{\mathsf{s}}^{\pi}(s) \rho'(s) \leq \mathcal{W}_{d_{\mathsf{s}}^{\pi}}(\rho, \rho') = \mathcal{W}_{d^{\pi}}(\rho, \rho'), \tag{43}$$

where the penultimate inequality arises from $(u, v) = \left( V_{\mathsf{s}}^{\pi}, V_{\mathsf{s}}^{\pi} \right)$ is a feasible solution to the Wasserstein distance $\mathcal{W}_{d_{\mathsf{s}}^{\pi}}(\rho, \rho')$ dual formulation in (19), and the last equality holds by (15).

**Proof of Lemma A.2** Before starting, we denote $p_{\alpha}$ as the probability that stopping at $d$ and $N$ as the stopping time. Then we consider the terms of interest in two cases. In the first case when $p = \frac{1}{2}$, invoking the facts in [65] directly leads to

$$N = D_{\mathsf{max}} d, \qquad p_{\alpha} = 1 - \frac{d}{d + D_{\mathsf{max}}}. \tag{44}$$

So the remainder of the proof will focus on the other case when $p \neq \frac{1}{2}$. Following the proof in [65], by basic calculus, it is easily verified that the following two centered terms are all martingales

$$S_n - (2p - 1)n, \qquad S_n^2 + 2(1 - 2p)S_n + (2p - 1)^2 n^2 + 4p(p - 1)n. \tag{45}$$

In continue, it can also be verified that

$$\mathbb{E}[S_N - (2p - 1)N] = p_{\alpha} d - (1 - p_{\alpha}) D_{\mathsf{max}} = 0, \tag{46}$$

which leads to

$$p_{\alpha} = \frac{D_{\mathsf{max}} + (2p - 1)N}{d + D_{\mathsf{max}}}. \tag{47}$$

Observing that

$$\mathbb{E}[S_n^2 + 2(1 - 2p)S_n + (2p - 1)^2 n^2 + 4p(p - 1)n] = 0 \tag{48}$$

yields

$$(2p - 1)^2 N^2 + [(D_{\mathsf{max}} - d)(1 - 2p) - (sp - 1)^2 - 1]N + D_{\mathsf{max}} d = 0, \tag{49}$$

and then implies

$$N \leq \sqrt{\frac{D_{\mathsf{max}}d}{1-2p}} \leq \sqrt{\frac{D_{\mathsf{max}}}{1-2p}}d. \tag{50}$$

Plugging in (50) into (47) leads to

$$p_\alpha \geq \frac{D_{\mathsf{max}} - \sqrt{D_{\mathsf{max}}d}}{d + D_{\mathsf{max}}} = 1 - \frac{\sqrt{D_{\mathsf{max}}d} + d}{d + D_{\mathsf{max}}} \geq 1 - \frac{2d\sqrt{D_{\mathsf{max}}}}{d + D_{\mathsf{max}}}, \tag{51}$$

which follows from (16) in Appendix A.2.

Finally, summing up the two cases, we arrive at

$$N \leq \max\left\{ D_{\mathsf{max}}d \,,\, \sqrt{\frac{D_{\mathsf{max}}}{(1-2p) \wedge 1}d} \right\}, p_\alpha \geq 1 - \frac{2d\sqrt{D_{\mathsf{max}}}}{d + D_{\mathsf{max}}}. \tag{52}$$

### A.6  Table of notations

We summarize useful notations in the proof here.

Table 1: Notations

| Symbol | Definition |
|---|---|
| $\Delta(X)$ | the set of probability measures on $X$ |
| $\mathsf{supp}(\rho)$ | support of distribution $\rho$ |
| $o^\pi(s; \rho)$ | $(1-\gamma)\sum_{t=0}^\infty \gamma^t \mathbb{P}\big(s_t = s \mid c \sim \rho; \pi\big)$ |
| $o^\pi(s, a; \rho)$ | $o^\pi(s; \rho)\pi(a \mid s)$ |
| $\mathcal{C}_{\mathsf{s}}(\rho, \pi)$ | $\{s : o^\pi(s; \rho) > 0\}$ |
| $\mathcal{C}(\rho, \pi)$ | $\{(s, a) : o^\pi(s, a; \rho) > 0\}$ |
| $V_{\mathsf{s}}^\pi(s)$ | $\mathbb{E}\big[\sum_{t=0}^\infty \gamma^t R(s_t, a_t, c') \mid s_0 = s, c = s; \pi\big]$ |
| $d_{\mathsf{s}}^\star$ | bisimulation distance $d_{\mathsf{s}}^{\pi^*}$ under the optimal policy $\pi^*$ |
| $a \wedge b$ | $\min(a, b)$ |

# B Experiment Details

In this section, we discuss details that could not be included in the main paper due to space limitations. This includes hyperaparameters of the algorithms, additional details about the environment, and visualizations that assist the qualitative analysis. The experiments were conducted on a desktop computer with an Intel Core i7-8700K CPU @ 3.70GHz 12-Core Processor, an Nvidia RTX 2080Ti graphics card and 64GB of RAM. All evaluation results are based on 30 episodes over 3 random seeds.

## B.1 Maze

We show the observation, action and context space in Table 2. The state, action and context space are discrete. For this toy-like example, we fix the maze layout throughout. The context is the initial position. The observation is the flattened value representation of the maze, including the goal, the current position of the agent, and the layout.

Table 2: Maze Environment Specifications

| Dim. | Discrete Observation Space |
|------|-----------------------------|
| 0-120 | Cell Type: $\{0 : free, 1 : wall, 2 : agent, 3 : goal\}$ |
| Index | Discrete Action Space |
| 0 | Go north |
| 1 | Go south |
| 2 | Go west |
| 3 | Go east |
| Index | Discrete Context Space |
| 0-50 | Initial position |

Table 3: Maze Hyperparameters

| PPO Learner | value |
|-------------|-------|
| gamma | 0.99 |
| learning_rate | 0.0001 |
| n_steps | 100 |
| ent_coef | 0.1 |
| total_timesteps | 100000 |
| GRADIENT hyperparameters | value |
| reward threshold $\bar{G}$ | -15 |

For this task, we use the PPO implementation in the `StableBaseline3` library [54]. We list the hyperparameters in the Table 3 (set to the default value if not mentioned in the table). We conduct hyperparameter grid search mainly on the following hyperparameters: $(learning\_rate, ent\_coef, n\_steps) \in \{0.0001, 0.0003, 0.001\} \times \{0.001, 0.01, 0.1\} \times \{10, 100, 200\}$

We visualize the intermediate task distributions generated by GRADIENT in Figure 8 with a $\Delta\alpha = 0.05$. For $\Delta\alpha = 0.1$ and $0.2$ are in the same figure as well (just with different gaps between two consecutive $\alpha$). The $\pi$-contextual-distance is computed using the A* path finding algorithm. Although it is not always realistic to have access to the optimal policy, this toy example is just to show the effectiveness of an appropriate contextual distance metric. More choices of the contextual distance could be explored in the future work.

## B.2 PointMass

In PointMass, the agent needs to navigate a pointmass through a wall with a small gap at an off-center position to reach the goal. The context is a 2-dimension vector representing the width and position of the gap on the wall. Following the setting in the original SPDL paper [8], the target distribution is an isotropic Gaussian distribution centered at $[2.5, 0.5]$ with a negligible variance $[4 \times 10^{-3}, 3.75 \times 10^{-3}]$ (which is effectively a point as shown in Figure 5a). The source distribution is an isotropic Gaussian distribution centered at $[0, 4.25]$ with a variance of $[2, 1.875]$. We show the observation, action and context space in Table 4.

For the baseline implantation, we use and modify the code base provided in [8]. We also use the best hyperparameters found by [8]. For ALP-GMM, they conduct hyperparameter grid search over $(p_{\text{RAND}}, n_{\text{ROLLOUT}}, s_{\text{BUFFER}}) \in \{0.1, 0.2, 0.3\} \times \{50, 100, 200\} \times \{500, 1000, 2000\}$. For Goal-GAN, they conduct grid search over $(\delta_{\text{NOISE}}, n_{\text{ROLLOUT}}, p_{\text{SUCCESS}}) \in \{0.025, 0.05, 0.1\} \times \{50, 100, 200\} \times \{0.1, 0.2, 0.3\}$. Due to unknown issue, the scale of episodic reward we get (about 0-100) is different from what is shown in [8] (0-10), nevertheless the trend of the training curves is

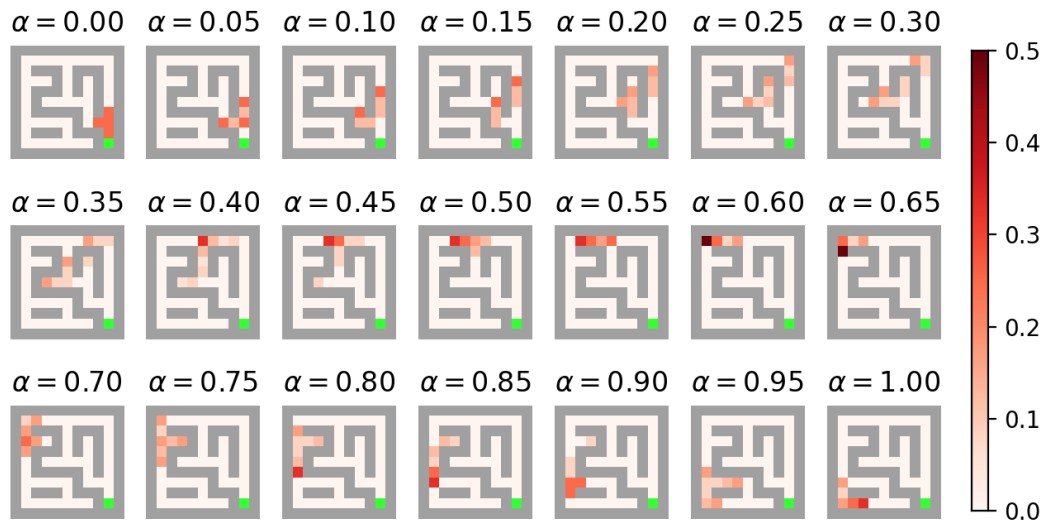

Figure 8: Intermediate task distributions generated by GRADIENT.

very similar. We visualize the context distribution of ALP-GMM and Goal-GAN in Figure 9. From this figure, we can reason why they underform GRADIENT and SPDL: since they cannot specify the target distribution, it is very difficult for them to learn to navigate through a narrow door at a specific position.

Table 4: PointMass Environment Specifications

| Dim. | Continuous Observation Space | range |
|---|---|---|
| 0 | x position | $[-4, 4]$ |
| 1 | x velocity | $[-\inf, \inf]$ |
| 2 | y position | $[-4, 4]$ |
| 3 | y velocity | $[-\inf, \inf]$ |
| **Dim.** | **Continuous Action Space** | |
| 0 | x force | $[-10, 10]$ |
| 1 | y force | $[-10, 10]$ |
| **Dim.** | **Continuous Context Space** | |
| 0 | gate position | $[-4, 4]$ |
| 1 | gate width | $[0.5, 8]$ |

## B.3 FetchPush

In FetchPush [58], the objective is to use the gripper to push the box to a goal position. The observation space is a 28-dimension vector, including information about the goal. The context is a 2-dimension vector representing the goal position on a surface. The target distribution is a uniform distribution over the circumference of a half-circle (Figure 5b). The source distribution is a uniform distribution over a square region centered at the box position, excluding the region within a certain radius of the object. We use this experiment to highlight the importance of the capability to handle arbitrary distributions rather than only the parametric Gaussian distributions. Since SPDL can only deal with parametric Gaussian distribution, we first fit the target and the source distribution with two Gaussian distributions and feed them into the baseline algorithms. The source Gaussian is $[1.14655655, 0.74819359]$ with a variance array of $[[0.0141083, 0.00055327], [0.00055327, 0.0149638]]$, and the target Gaussian is $[1.33561676, 0.74819359]$ with a variance array of $[[8.89519060e-03, -1.34467507e-18], [-1.34467507e-18, 4.59090909e-02]]$. We visualize the context distribution of ALP-GMM and Goal-GAN in Figure 10

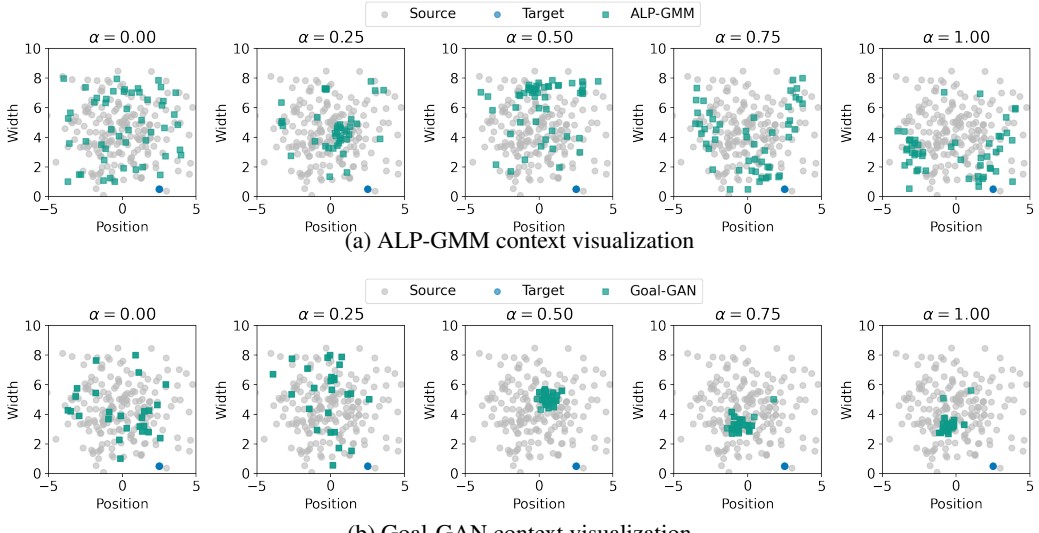

(a) ALP-GMM context visualization

(b) Goal-GAN context visualization

Figure 9: Visualizations of context distributions and curricula in PointMass. The contexts are taken from environment steps at 10k, 50k, 100k, 200k, and 300k.

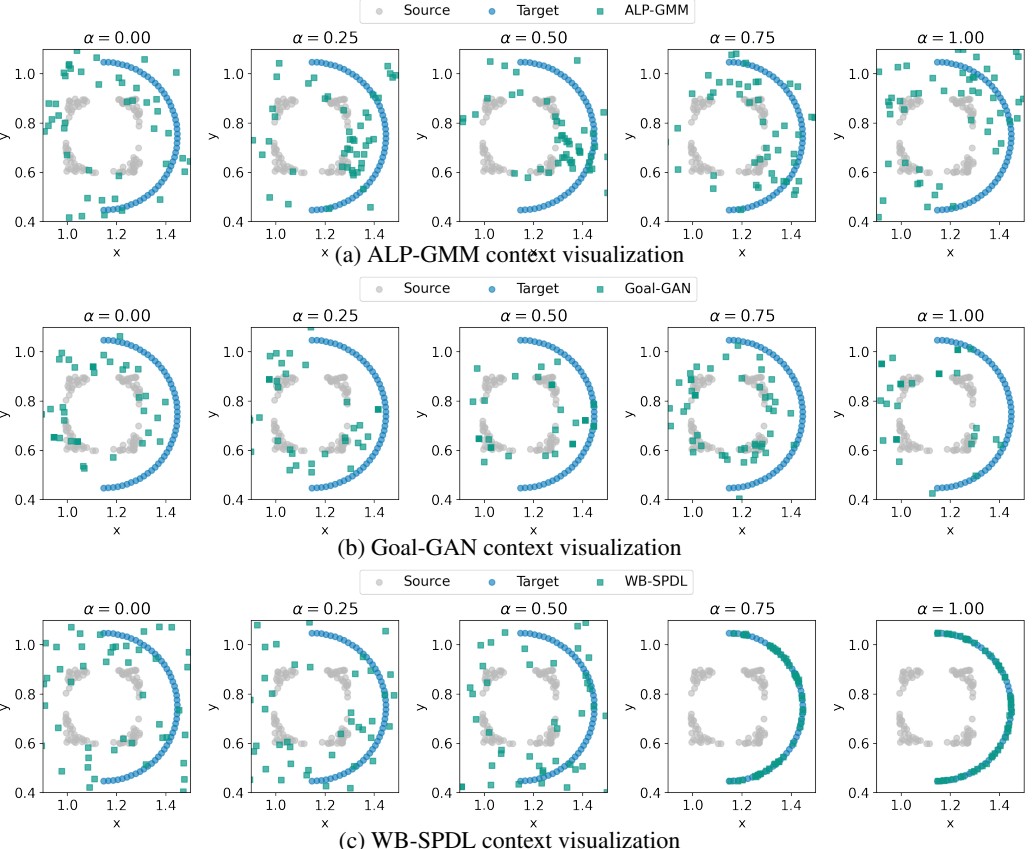

(a) ALP-GMM context visualization

(b) Goal-GAN context visualization

(c) WB-SPDL context visualization

Figure 10: Visualizations of context distributions and curricula in FetchPush. The contexts are taken from environment steps at 5k, 50k, 100k, 250k, and 500k for ALP-GMM and Goal-GAN, 100, 500, 1000, 1200, 1500 for WB-SPDL.

Table 5: PointMass Hyperparameters

| SAC Learner | value |
| --- | --- |
| train_freq | 5 |
| buffer_size | 10000 |
| gamma | 0.95 |
| learning_rate | 0.0003 |
| learning_starts | 500 |
| batch_size | 64 |
| net architecture | [64, 64] |
| activation_fn | Tanh |

| SPDL hyperparameter | value |
| --- | --- |
| $\alpha$ offset | 25 iterations |
| $\zeta$ | 1.1 |
| KL threshold | 8000 |
| maximum KL | 0.05 |
| steps per iteration | 2048 |

| ALP-GMM hyperparameter | value |
| --- | --- |
| random task ratio | 0.2 |
| fit rate (number of episodes between two fit of GMM) | 200 |
| max size (maximal number of episodes for computing ALP) | 1000 |

| GOAL-GAN hyperparameter | value |
| --- | --- |
| state noise level | 0.05 |
| fit rate (number of episodes between two fit of GAN) | 25 |
| probability to sample state with noise | 0.05 |

| GRADIENT hyperparameter | value |
| --- | --- |
| $\Delta\alpha$ | 0.1 |
| Reward threshold $\bar{G}$ | 40 |

Table 6: FetchPush Environment Specifications

| Dim. | Continuous Observation Space | range |
| --- | --- | --- |
| 0-2 | grip_pos | $[-\inf, \inf]$ |
| 3-5 | object_pos | $[-\inf, \inf]$ |
| 6-8 | object_rel_pos | $[-\inf, \inf]$ |
| 9-10 | gripper_state | $[-\inf, \inf]$ |
| 11-13 | object_rot | $[-\inf, \inf]$ |
| 14-16 | object_velp | $[-\inf, \inf]$ |
| 17-19 | object_velr | $[-\inf, \inf]$ |
| 20-22 | grip_velp | $[-\inf, \inf]$ |
| 23-24 | gripper_vel | $[-\inf, \inf]$ |
| 25-27 | goal_pos | $[-\inf, \inf]$ |

| Index | Continuous Action Space | |
| --- | --- | --- |
| 0-2 | pos_ctrl | $[-1, 1]$ |
| 3 | gripper_ctrl | $[-1, 1]$ |

| Dim. | Continuous Context Space | |
| --- | --- | --- |
| 0 | goal_x_pos | $[0.8, 1.5]$ |
| 1 | goal_y_pos | $[0.4, 1.1]$ |

Table 7: FetchPush Hyperparameters

| SAC Learner | value |
|---|---|
| train_freq | 1 |
| buffer_size | 100000 |
| gamma | 0.99 |
| learning_rate | 0.001 |
| learning_starts | 1000 |
| batch_size | 256 |
| net architecture | [64, 64, 64] |
| activation_fn | Tanh |
| SPDL hyperparameter | value |
| $\alpha$ offset | 0 iterations |
| $\zeta$ | 1.0 |
| KL threshold | 20 |
| maximum KL | 0.05 |
| steps per iteration | 5000 |
| ALP-GMM hyperparameter | value |
| random task ratio | 0.3 |
| fit rate (number of episodes between two fit of GMM) | 200 |
| max size (maximal number of episodes for computing ALP) | 1000 |
| GOAL-GAN hyperparameter | value |
| state noise level | 0.1 |
| fit rate (number of episodes between two fit of GAN) | 200 |
| probability to sample state with noise | 0.3 |
| GRADIENT hyperparameter | value |
| $\Delta\alpha$ | 0.2 |
| Reward threshold $\bar{G}$ | -25 |

# C Algorithm Details

## C.1 Exact Computation of $\pi$-contextual-distance for Maze

For the maze, we compute the exact $\pi$-contextual-distance using dynamic programming. We iteratively compute the $\pi$-contextual-distance (represented by a matrix) until the maximum difference in metric estimate between successive iterations is smaller than a tolerance. Due the Assumption 4.1, the computation of $\pi$-contextual-distance is largely simplified (since the context space is essentially overlapped with the state space). We present the pseudocode as follows:

---

**Algorithm 3:** Compute the exact metric when $\mathcal{S} = \mathcal{C}$

---

**Input:** environment $env$, context space size $n$, tolerance $\epsilon_{tol}$, discounting factor $\gamma$, agent policy $\pi$

$\mathbf{M} \leftarrow \mathbf{0}_{n \times n}$;// initialize with zero matrix of $n$ by $n$
$\delta_{\mathbf{M}} = 2\epsilon_{tol}$;
**while** $\delta_{\mathbf{M}} > \epsilon_{tol}$ **do**
    $\mathbf{M}' \leftarrow \mathbf{0}_{n \times n}$;
    **for** $s_1$ *in* $0, 1, 2, \ldots, n-1$ **do**
        **for** $s_2$ *in* $0, 1, 2, \ldots, n-1$ **do**
            $a_1 \leftarrow \pi(s_1), a_2 \leftarrow \pi(s_2)$;
            $s_1', r_1 \leftarrow env.step(s_1, a_1)$
            $s_2', r_2 \leftarrow env.step(s_2, a_2)$
            $\mathbf{M}'[s_1, s_2] \leftarrow |r_1 - r_2| + \gamma \mathbf{M}[s_1', s_2']$
    $\delta_{\mathbf{M}} \leftarrow \max |\mathbf{M}' - \mathbf{M}|$
    $\mathbf{M} \leftarrow \mathbf{M}'$
Add small offset to the diagonal terms of $\mathbf{M}$ to avoid computation issues;
$\mathbf{M} \leftarrow \frac{\mathbf{M}}{\max \mathbf{M}}$// normalize to have the maximum value of 1
**Output:** Contextual Distance Metric $\mathbf{M}$

---

## C.2 Learning Embeddings to Encode Non-Euclidean Distance

Due to the computation complexity of the free-support Wasserstein barycenters, it is generally difficult to compute them, especially for non-euclidean cost metric. There are some attempts to use neural networks and stochastic gradient descent to solve for the barycenters approximately [60]. Another possible route is to learn embeddings such that the euclidean interpolation in the latent space assembles the interpolation in the non-euclidean original space. In the fixed-support setting, Deep Wasserstein Embedding (DWE) [61] uses siamese networks to learn an latent space where the euclidean distance approximates the Wasserstein distance in the original space. We could adopt a similar method but for the free-support, i.e., continuous context space.

Let the encoder be $\epsilon(\cdot)$ and decoder be $\delta(\cdot)$, given pairs of contexts $\{c_1^i, c_2^i\}_{i=1,\ldots,N}$ and their contextual distance $\{d(c_1^i, c_2^i)\}_{i=1,\ldots,N}$, the global objective funtion is to minimize

$$\min_{\epsilon, \delta} \sum_i \left\| \|\epsilon(c_1^i) - \epsilon(c_2^i)\|^2 - d(c_1^i, c_2^i) \right\|^2 + \lambda \sum_{k=1,2} \|\delta(\epsilon(c_k^i)) - c_k^i\|^2 \tag{53}$$

The first term is the loss for distance embedding, and the second term is for reconstruction. Other regularization could be added to improve the robustness. After obtaining the trained encoder and decoder, we can first encode the source and target contexts to the latent space, compute the Wasserstein barycenter in the latent space and finally decode the latent barycenter back to the original context space. The main algorithm GRADIENT with distance embedding is shown in Algorithm 4.

**Algorithm 4:** GRADIENT with Distance Embedding

---

**Input:** Source task distribution $\mu(c)$, target task distribution $\nu(c)$, interpolation factor $\Delta\alpha$, reward threshold $\bar{G}$.

Initialize the agent policy $\pi$;

$\alpha \leftarrow 0$;

**for** $k$ *in* $0, 1, 2, ..., K$ **do**

    **if** $k == 0$ **then**

        |  $\rho(c) \leftarrow \mu(c)$;

    **else**

        |  $\rho(c) \leftarrow \delta(\texttt{ComputeBarycenter}(\epsilon(\mu), \epsilon(\nu), \alpha, l_2))$;

    $\{c_i, R_i\}_{i=1,...,M} \leftarrow$ Optimize $\pi$ in the task distribution $\rho(c)$ until $G > \bar{G}$ (potentially with some exploration noise for $c$), and return recent $M$ sampled context $c_i$ and corresponding episodic rewards $R_i$;

    Estimate $J^\pi(c)$ from $\{c_i, R_i\}_{i=1,...,M}$ using Gaussian Process;

    `// Use the absolute difference between episodic reward to define the`
    `   distance metric as an example`

    Define $d^\pi(c_i, c_j) := |J^\pi(c_i) - J^\pi(c_j)|$;

    Encoder $\epsilon$, Decoder $\delta \leftarrow \texttt{EmbedDistanceMetric}(d^\pi)$; `// Algorithm 5`

    $\alpha \leftarrow \alpha + \Delta\alpha$;

    $\mu(c) \leftarrow \rho(c)$;

**Output:** Agent policy $\pi$

---

**Algorithm 5:** EmbedDistanceMetric

---

**Input:** contextual distance metric $d$.

Initialize encoder $\epsilon$ and decoder $\delta$;

Uniformly sample pairs of contexts from the context range $\{c_1^i, c_2^i\}_{i=1,...,N}$; `// This step is`
`   cheap since there is no interaction with environment required.`

Compute $\{d(c_1^i, c_2^i)\}_{i=1,...,N}$;

Train encoder and decoder by minimizing (53);

**Output:** Encoder $\epsilon$, Decoder $\delta$

---

We show an example of using Algorithm 4. Here we consider a classical U-shaped maze with continuous state, action and context space. We assume that the source and target distributions are two Gaussian distributions at the two ends of the maze. Due to the existence of the obstacle in the middle, it is not appropriate to use the $l_2$ distance as the contextual distance. In this case, we use $d^\pi(c_1, c_2) := |J^\pi(c_1) - J^\pi(c_2)|$.

The interpolation results are shown in Figure 11. At the first a few stages, since the agent does not have a good policy as well as a good estimate of the episodic reward, the interpolation results are not very good. However, since $\alpha$ is small, the barycenters are relatively close to the source distribution, so GRADIENT does not generate too unreasonable tasks. With the learning progressing, the estimates become better and better and therefore produces much better interpolation results.

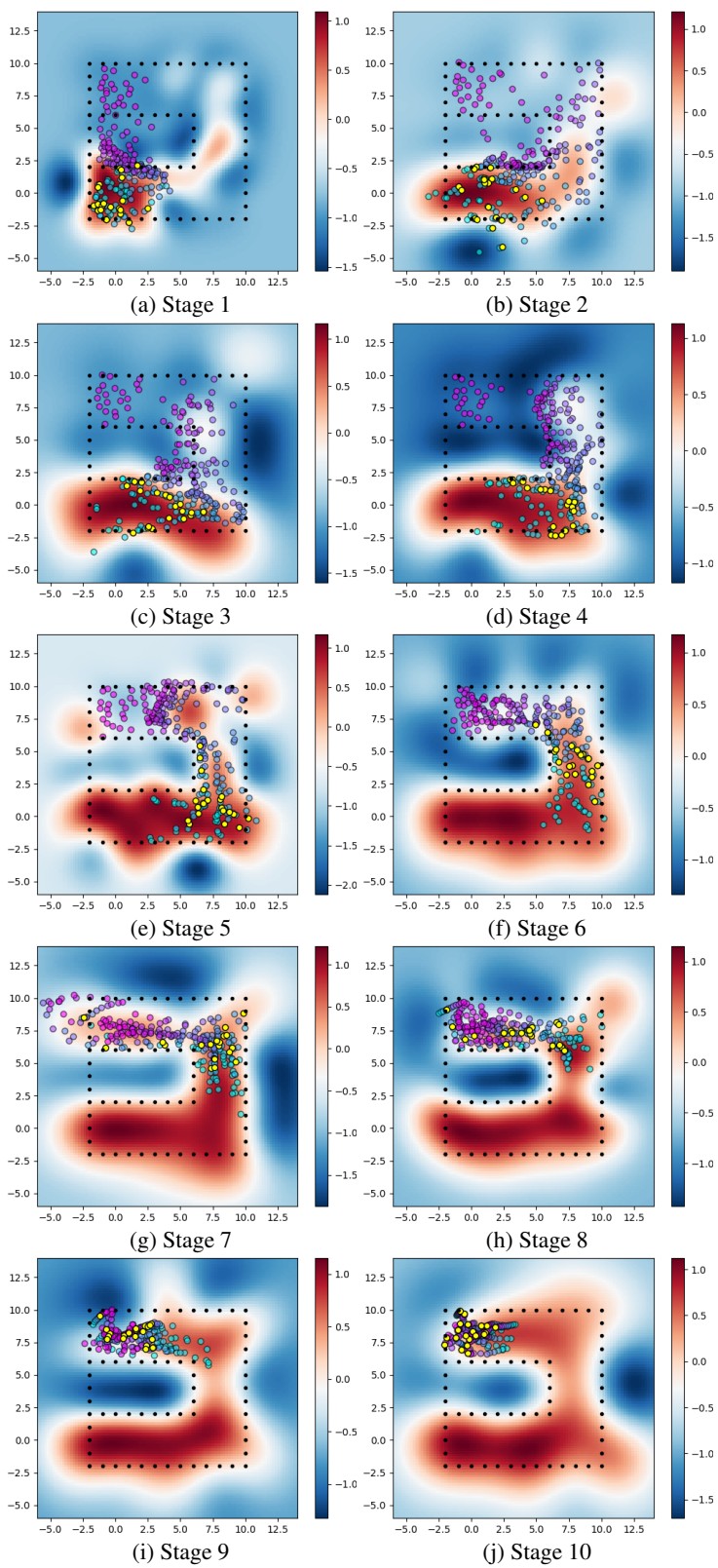

Figure 11: U-Maze interpolation results. The original source and target distributions are two Gaussian centered at $[0, 0]$ and $[0, 8]$. The color of the heat map represents the estimate of $J^\pi(c)$. The colored circle represent the Wasserstein interpolation. More specifically, the circles with cyan to purple color represent the interpolation results from the current source to target with $\alpha = [0, 1]$. The yellow circles highlight the barycenter corresponding to the current stage.