# OpenReview forum: "Curriculum Reinforcement Learning using Optimal Transport via Gradual Domain Adaptation"
_NeurIPS.cc/2022/Conference — NeurIPS 2022 Accept_

### Official Review · Reviewer_4tAz · 2022-06-19

**Rating:** 7
**Confidence:** 3
**Soundness:** 3 good
**Presentation:** 4 excellent
**Contribution:** 3 good

**Summary:**

This work proposes GRADIENT, a method for curriculum learning, by interpolating the initial context distribution and the target context distribution using the Wasserstein distance. This paper provides a theoretical justification for the proposed algorithm that the optimal value function under two adjacent contexts can be bounded by the Wasserstein distance between the contextual distributions of these adjacent contexts. This paper also demonstrates the practicality of the GRADIENT method in several goal-condition tasks.

**Questions:**

1. It seems that the context dimension of all experiments conducted so far is either discrete or dimensional <=2, is it possible to run the GRADIENT algorithm on some non-goal-conditioned tasks?
2. It seems that assumption 4.1 assumes the state space is a subset of the context space ($\mathcal{S}\subset\mathcal{C}$), but in the FetchPush environment, the context is a 2-dimensional vector representing the goal while the state is a 28-dimensional vector, perhaps the author can clarify this?

**Strengths And Weaknesses:**

Pros:
1. The main body of this paper is very well written and the Wasserstein distance indeed serves good metric for measuring distributions.
2. The experimental results are very interesting as they clearly reflect the interpolation of the Wasserstein Barycenter.
3. The experimental results seem promising -- as in all demonstrated tasks, they outperform the prior arts by a large margin.

Cons (Updated Aug 3rd):
1. (Major; clarified) The connection between the theoretical results and the experiments is not carefully addressed. The paper would be much better if the author can address how the main assumption (4.1) is reflected in the experiments. Since all experiments are low-dimensional, so it should not incur the "curse of dimensionality of the Wasserstein distance", hence it would be nice to address how the assumption is actually reflected in the selected experiments.
2. (Major; draft improved) The proof of theorem 4.1 is hard to follow. The readability of the paper would be largely improved if the author can summarize the notations/assumptions in the appendix, as the proof adopt many non-standard RL theory notations.
3. (Minor; draft improved) There are some minor citation issues in the related work section -- when using a paper to start a sentence, it would be better if the author used \citet{} instead of \citep{} (e.g, "[6] proposes Self-PacedReinforcementLearning(SPRL)"; "Concurrent to our work,[19] also propose to replace KL divergence").

The reviewer would like to improve the rating if the author can improve the readability of the proof.

---

> ### Author Response · Authors · 2022-08-02
> **Response from authors, 2/2**
>
> > It seems that the context dimension of all experiments conducted so far is either discrete or dimensional <=2, is it possible to run the GRADIENT algorithm on some non-goal-conditioned tasks?
>
> GRADIENT is capable of handling contextual MDPs, which include goal-conditioned tasks as a subset, as we mentioned in Preliminary Section 3.1 (Line 115). Therefore, GRADIENT is able to run on non-goal-conditioned tasks. Moreover, it is worth mentioning that not all the experiments we conducted are strictly-speaking goal-conditioned. For example, for the maze environment, we do not changeset the goal but instead, vary the initial position. For the pointmass environment, we control the width and position of the gap.
> We agree that contexts now in our experiments are either discrete or low-dimensional. Such experiment design choice is common in related works and baselines, such as [1, 2, 3]. It comes from both the fact of many tasks can be represented by a low-dimensional vector, such as goal positioned or physical parameters of robots, and the consideration of easier visualization. We believe how to deal with high-dimensional contexts, such as images, is indeed a very promising future direction. One idea is to use a variational autoencoder to map the high-dimensional data to low-dimensional latent space, and then compute the Wasserstein barycenter in the latent space. This idea has been explored in some RL representation learning literature [4, 5].
>
> **Reference:**
>
> [1] Pascal Klink, Hany Abdulsamad, Boris Belousov, Carlo D’Eramo, Jan Peters, and Joni Pajarinen. A probabilistic interpretation of self-paced learning with applications to reinforcement learning. Journal of Machine Learning Research, 22(182):1–52, 2021.
>
> [2] Carlos Florensa, David Held, Xinyang Geng, and Pieter Abbeel. Automatic goal generation for reinforcement learning agents. In International conference on machine learning, pages 1515–1528. PMLR, 2018.
>
> [3] Rémy Portelas, Cédric Colas, Katja Hofmann, and Pierre-Yves Oudeyer. Teacher algorithms for curriculum learning of deep rl in continuously parameterized environments. In Conference on Robot Learning, pages 835–853. PMLR, 2020.
>
> [4] Zhang, Amy, Rowan McAllister, Roberto Calandra, Yarin Gal, and Sergey Levine. Learning invariant representations for reinforcement learning without reconstruction. arXiv preprint arXiv:2006.10742 (2020).
>
> [5] Hansen-Estruch, Philippe, Amy Zhang, Ashvin Nair, Patrick Yin, and Sergey Levine. Bisimulation Makes Analogies in Goal-Conditioned Reinforcement Learning. arXiv preprint arXiv:2204.13060 (2022).

---

> > ### Comment · Reviewer_4tAz · 2022-08-03
> > **Thanks for the clarification**
> >
> > Thank you very much for the clarification. The reviewer believes the work could better appeal to both theory and practice community if the author can also discuss this issues in future updates.

---

> ### Author Response · Authors · 2022-08-02
> **Response from authors, 1/2**
>
> Thank you for the reviews. We are glad you find our paper well-written and our experiments interesting and promising. We hope we can address your concerns about the assumptions and the readability of the proof.
>
> > The connection between the theoretical results and the experiments is not carefully addressed...
>
> > It seems that assumption 4.1 assumes the state space is a subset of the context space...
>
> In order to understand how properties of context distributions affect the curriculums needed for domain adaptation, we study a rigorous case where an environment has discrete state/action/context spaces, and the context only affects the initial state distribution (Assumption 4.1), which is exactly the setting of the discrete maze environment. In this environment, the context space coincides with the state space and dictates the initial state distribution (there is a typo in the original version: $\subset \rightarrow \subseteq$).
> While this setting is clearly much simpler than some contextual RL environments currently being used (Pointmass and FetchPush), the simplicity of our theoretic models is a strength because we established the guarantee of the performance gap during the domain adaptation process. Hence no amount of algorithmic ingenuity can neglect the benefit brought by our GDA approach. We believe our main theorem is a significant first step toward a theoretical understanding of GDA in CRL domain.
>
> > The proof of theorem 4.1 is hard to follow. The readability of the paper would be largely improved if the author can summarize the notations/assumptions in the appendix, as the proof adopt many non-standard RL theory notations. The reviewer would like to improve the rating if the author can improve the readability of the proof.
>
> Thanks to the reviewer for the helpful comments and suggestions on the readability of the proof! We update the entire presentation of the proof in the appendix and include it in the new version, including but not limited to rewriting the whole presentation and clarifying important notations more clearly. We hope that this makes the proof clear for the reviewer to read. And we really appreciate and welcome further comments. Here, we summarize the main updates of the proof in Appendix A on a point-by-point basis.
> 1. We add a paragraph at the beginning of Appendix A.1 for the convenience of the reader to have a glance at the organization before getting into further details.
> 2. For convenience, we summarize all the useful notations introduced in the main text and the proof in Appendix A.6.
> 3. We summarize all the key properties used in the proof in a separate subsection Appendix A.3. We hope that this can lead to a more clear structure of the proof.
> 4. Combing through the proof, we rewrite the entire presentation to show the proof of the main Theorem. We reorganized the additional notations and definitions in Appendix A.1 by adding more bullet points and explanations. In addition, to clarify assumptions for the reader, we summarize the additional assumptions in Appendix A.2 and give more explanations.
>
> Given the improvements in readability, would you consider increasing your support for our work to "Accept"?

---

> > ### Comment · Reviewer_4tAz · 2022-08-03
> > **Thanks for the update**
> >
> > Dear Authors,
> >
> > Thank you very much for your replies and the updated draft. I think the updated proof of thm 4.1 now is clear and easy to follow. I have increased my rating based on the updated proof, but still, I do believe in the future the distinctions between theory and experiments should be addressed better.

---

> > > ### Author Response · Authors · 2022-08-09
> > > **Re:**
> > >
> > > Thank you very much for the discussions! We are glad that the presentation of the proof has been improved. We will keep polishing the final version according to your suggestions.

---

### Official Review · Reviewer_KSyL · 2022-06-20

**Rating:** 6
**Confidence:** 2
**Soundness:** 3 good
**Presentation:** 3 good
**Contribution:** 3 good

**Summary:**

this paper propose GRADIENT, a novel curriculum reinforcement learning method based on optimal transport. Therefore, GRADIENT improves the learning efficiency and asymptotic performance in a wide range of tasks compared with state-of-the-art baselines.

For example, in the maze example, GRADIENT algorithm can better interpolates the environment task distribution from source to target env compared to linear interpolation.

**Questions:**

Q: dealing with sparse rewards problem, there are some other useful RL algorithms, such as HER. have the authors conducted some experiments compared to these baselines? and why GRADIENT?

I appreciate the authors add HER as a baseline. Considering the discussion and revision, I increased my score

**Strengths And Weaknesses:**

Strengths:

OT in CRL is novel

the proposed method deals with both discrete and continuous context space.

empircally,this paper shows its proposed method's learning efficiency

this paper provides a theorectical bound for policy transfer.

---

> ### Author Response · Authors · 2022-08-02
> **Response from authors**
>
> Thank you for the review. We are glad you agree that our method is novel and the capability to deal with both discrete and continuous context is a strength. Given your overall positive view of this work, we hope we can address your interests around HER baseline and that you will consider increasing your rating of our work.
>
> > Dealing with sparse rewards problem, there are some other useful RL algorithms, such as HER. have the authors conducted some experiments compared to these baselines? and why GRADIENT?
>
> We have added HER baseline and updated Figure 4b. We observe that our method still outperforms HER. The potential reason is that HER relies on a random sampling of hindsight goals which is very inefficient in the later stage of training and does not consider the existence of a target goal distribution (therefore the implicit curriculum will not move towards such a target distribution). This also highlights the need for a target-aware curriculum design.
>
> It is worth pointing out the difference in problem settings between HER and the methods we considered in this paper:
> * HER produces implicit curricula by relabelling the goal and reward. It relies on a reply buffer and therefore is only applicable to the off-policy learners. However, GRADIENT produces explicit curricula by directly generating contexts, which does not require a reply buffer and therefore works for both on- and off- policy learners. Other baselines we compared could have similar considerations therefore do not include HER for fair comparison in the paper either [1-3].
> * HER is only applicable in goal-conditioned MDP, which is a subset of contextual MDP). However, GRADIENT can be applied to more general contextual MDP settings (see maze and pointmass experiments).
>
> With the HER baseline added and reasoning discussed, could the reviewer reconsider the evaluation and increase the rating to accept?
>
> **Reference:**
>
> [1] Pascal Klink, Hany Abdulsamad, Boris Belousov, Carlo D’Eramo, Jan Peters, and Joni Pajarinen. A probabilistic interpretation of self-paced learning with applications to reinforcement learning. Journal of Machine Learning Research, 22(182):1–52, 2021.
>
> [2] Carlos Florensa, David Held, Xinyang Geng, and Pieter Abbeel. Automatic goal generation for reinforcement learning agents. In International conference on machine learning, pages 1515–1528. PMLR, 2018.
>
> [3] Rémy Portelas, Cédric Colas, Katja Hofmann, and Pierre-Yves Oudeyer. Teacher algorithms for curriculum learning of deep rl in continuously parameterized environments. In Conference on Robot Learning, pages 835–853. PMLR, 2020.

---

> > ### Author Response · Authors · 2022-08-09
> > **Re:**
> >
> > Thank you very much for the prompt response and raising the rating! We are glad that we have addressed your concerns.

---

### Official Review · Reviewer_pAPM · 2022-07-08

**Rating:** 6
**Confidence:** 3
**Soundness:** 2 fair
**Presentation:** 3 good
**Contribution:** 2 fair

**Summary:**

The paper considers the problem of building curricula of tasks for reinforcement learning in order to learn the target task more easily. The authors propose to approach this problem by finding the interpolations between tasks using the Optimal Transport framework, thus replacing previous methods which heavily relied on f-divergences and the KL divergence in particular. Additionally, the paper introduces the $\pi$-contextual-distance in order to measure the distance between different contexts, particularly in settings where the context space is discrete. The authors provide theoretical proof that under certain assumptions the performance gap between each pair of neighboring tasks is small. Finally, they test the proposed method empirically and show that it outperforms competitors in settings where the context cannot be well approximated with a Gaussian distribution.

**Questions:**

The only suggestion I think is reasonable within the scope of the rebuttal period is adding the limitations to the paper. Other than that, please refer to the "Weaknesses" and "various comments" sections above.

**Limitations:**

The paper does not discuss the potential negative societal impact and I don't think such discussion is required.

There is no discussion on the limitations of this work, which I think would be very useful for the reader. I listed a few possible limitations in the  Weaknesses section above.

**Strengths And Weaknesses:**

Strengths:
- The proposed approach is intuitive and follows naturally from the previous line of research on applying KL divergence to find a good sequence of tasks.
- The authors provide interesting and plausible scenarios when their approach is especially useful. In particular, Figure 5 illustrates this point visually by plotting the context space.
- The results in settings where the context cannot be effectively modeled as a Gaussian distribution are quite good.
- The paper is well written, the description of the method and experiments is mostly clear, and the mathematical notation is well defined.

Weaknesses:
- The empirical evaluation could be extended, as two out of three tasks (Maze and PointMass) are rather toyish. It would be useful to have another more complicated benchmark, such as the Ant locomotion environment that was used e.g. in the SPDL paper.
- Although the checklist at the end of the paper says otherwise, the limitations are not properly described. For example, I would suggest discussing the following limitations:
    - The method is only tested in settings where the context space is nice and relatively low-dimensional.
    - The assumptions for Theorem 4.1 are quite strict (the context cannot change the transition dynamics nor the reward function).
    - The $\Delta\alpha$ hyperparameter seems to be very important, but there is no practical guidance on how to set it.
- This is subjective, but I'm not entirely convinced about the level of novelty of the proposed approach. The main idea is to take an approach that works well with the Kullback-Leibler divergence and use another way to measure the similarity of distributions, namely the Wasserstein distance. In the same way, the $\pi$-contextual-distance which the authors list as a contribution seems to be a straightforward extension of the distance introduced in [1]. On the one hand, this seems a bit iterative, but on the other I still think overall the contribution is significant, so treat this as minor criticism.

In the end, although I have some doubts about novelty and empirical evaluation, I think the contributions in this paper are significant. For the moment I decided to go with "Weak accept".

Various comments:
Figure 3(b) (and the rest of the figures with performance): Just to make sure, does the x-axis shows the number of steps of training on the target environment or the whole training process (i.e. starting with pre-training on the auxiliary task)?
Line 111: unnecessary parenthesis.

[1] Castro, Pablo Samuel. “Scalable methods for computing state similarity in deterministic Markov Decision Processes.” AAAI (2020).

---

> ### Author Response · Authors · 2022-08-02
> **Response from authors**
>
> Thank you for the review and suggestions. We are glad that you find our approach intuitive, the experiments interesting, and our paper well-written. Given your positive view of this work, we hope we can address your concerns around the novelty of our method and limitations, and that you will consider increasing your rating of our work.
>
> > The empirical evaluation could be extended, as two out of three tasks (Maze and PointMass) are rather toyish.
>
> Thanks again for suggesting a possible extension to the experiment design. We want to highlight the fact that the purpose of our experiments was to examine the important aspects of curriculum design such as the ability to handle non-parametric distribution. Therefore we would like to focus on the curriculum generated rather than the learning task complexity by itself, as we use the same learner for all the methods. Although Ant locomotion could be a more complex control task by itself, the context space is still a continuous 2-dimensional vector space, which leads to a similar level of difficulty for the curriculum designer as in our experiments. We do agree that extending the method to handle extremely high-dimensional context space is a very interesting future direction.
>
> > The only suggestion I think is reasonable within the scope of the rebuttal period is adding the limitations to the paper.
>
> Thank the reviewer for suggesting a dedicated section for the limitations. We have added it towards the end of the paper and paste it here:
> ### Limitations
> In our experiments, we examine the settings that are either discrete or low-dimensional. Such experiment design choice is common in the related works and baselines [11, 10, 6]. It comes from both the fact of many tasks can be represented by a low-dimensional vector, such as goal positioned or physical parameters of robots, and the consideration of easier visualization. We believe how to deal with high-dimensional contexts, such as images, is a very promising future direction. In addition, our analysis is conducted based on a restrictive subset of contextual MDPs (Assumption 4.1) where the context controls only the initial state distribution. It would be beneficial to relax this assumption by imposing assumptions on the specific manner in which the context controls the dynamics or reward. Nevertheless, we believe that our main theorem is a significant first step toward a theoretical understanding of GDA in CRL domain.
>
> Regarding the hyperparameter $\Delta \alpha$, we conducted a preliminary sensitivity analysis in Section 5.1. The main finding is that $\Delta \alpha$ could balance the trade-off between how fast the curriculum progress towards the target and how much the performance drop the agent could experience.
>
>
> > This is subjective, but I'm not entirely convinced about the level of novelty of the proposed approach...but on the other I still think overall the contribution is significant, so treat this as minor criticism.
>
> Thanks to the reviewer for acknowledging that our overall contribution is significant. We would like to expand a little more discussion on the novelty of GRADIENT in terms of the methodology. First of all, Wasserstein distance enables us to encode rich metric information when measuring the distance between distributions, which is impossible via KL divergence. Moreover, Wasserstein distance can deal with those distributions that don’t share the same support, which is impossible to measure with KL divergence.
> The metric function we proposed, namely $\pi$-contextual-distance, together with the Wasserstein geodesics, establishes a principled approach to curriculum reinforcement learning. Last but not least, GRADIENT bridges state-of-the-art techniques in GDA and CRL in a novel and valuable way, which could open the door for more analysis and understanding of curriculum reinforcement learning.

---

> > ### Comment · Reviewer_pAPM · 2022-08-05
> > **Response to the authors**
> >
> > Thank you for the thorough response. I appreciate the answers and in particular adding discussing limitations in the paper. For the moment, I decided not to change the score (weak accept).
> >
> > **Additional experiments** -- I understand your motivation, but I still think a more thorough empirical investigation would improve the paper. The effect of the curriculum might be different when the underyling control task is more complex.
> >
> > **Novelty** -- I definitely agree that the Wasserstein distance is in many situations much more useful than f-divergences, but it's the idea of replacing one kind of metric with another has limited novelty from my perspective. Still, this is a minor point and as I mentioned, I think the contribution is overall significant.
> >
> > **Prior work** -- Other reviewers rightly pointed out that this paper shares certain similarities with [1]. In addition, I would also like to highlight [2], which might be a more refined version of [1] that I found after writing the original review. Given that this paper was, to the best of my knowledge, published after the NeurIPS paper submission deadline, it is a concurrent work and the authors shouldn't be expected to compare with this paper. As such, I also do not take that previous paper into consideration when deciding the score. However, I still think that the community would benefit from positioning this paper wrt the prior work in the related work section of the paper.
> >
> > [1] Klink et al. Metrics Matter: A Closer Look on Self-Paced Reinforcement Learning. 2021
> >
> > [2] Klink, Pascal, et al. "Curriculum Reinforcement Learning via Constrained Optimal Transport." International Conference on Machine Learning. PMLR, 2022.

---

> > > ### Author Response · Authors · 2022-08-09
> > > **Re:**
> > >
> > > Thank you very much for the suggestions on the experiments and the related work section. Indeed we also notice [2] which is an updated version of [1]. In the discussion with reviewer G5ua and the updated version, we pointed out several important aspects of the difference between our work and [1]. We will update the related work section to better position our paper w.r.t. the community. We are very grateful for all the valuable discussion!

---

### Official Review · Reviewer_G5ua · 2022-07-09

**Rating:** 5
**Confidence:** 4
**Soundness:** 2 fair
**Presentation:** 2 fair
**Contribution:** 2 fair

**Summary:**

This paper considers the task of curriculum reinforcement learning (CRL), and propose an algorithm, GRADIENT, which leverages ideas from gradual domain adaptation (GDA) and optimal transport (OT). Essentially, GRADIENT focuses on the probability distribution $\rho(c)$ over the context variable $c$, and fine-tunes it from the source distribution to the target distribution gradually by treating it as a Wasserstein barycenter between the source and target (moving gradually from the source to target). The authors performed some brief theoretical analysis and conducted experiments over several benchmarks. The empirical results show GRADIENT has better sample complexity compared with the considered baselines.

**Questions:**

See Weaknesses above.

**Limitations:**

See Weaknesses above.

**Strengths And Weaknesses:**

Strengths:

1. The paper applies gradual domain adaptation (GDA) to reinforcement learning in a reasonable way. It seems that the task of curriculum reinforcement learning (CRL) involves a directional adaptation (i.e., from a source to a target) of probability distribution in a continuous way , which is indeed what GDA could help with. It is good to see that GDA is becoming useful in more and more application scenarios.

2. The authors provide a theoretical analysis of the proposed algorithm to corroborate it, which is nice. It looks that the derived RL bounds, Eq. (6)(8), look very similar to the bound in Lemma 1 of [1], which are derived in the supervised learning setting for GDA. Overall, I think the theoretical analysis is elegant. (Notice to area chair: I didn't check the proof since I'm not an expert of RL theory)

Weaknesses:

1. Novelty: The application of GDA and Wasserstein barycenter is quite straightforward, as long as one knows both GDA and CRL. So from the perspective of GDA, this paper is not very novel in terms of methodology.

2. [**IMPORTANT**] Missing discussions on a concurrent/prior work: This paper briefly mentioned a concurrent work [2] in Sec. 2 in one sentence. However, [2] actually has lots of similarity with this work, and it appeared online in Oct 2021, which seems more like a prior work rather a concurrent one. [2] also applied optimal transport (OT) and Wasserstein barycenters to CRL, thus in my opinion, it should be discussed in this paper more carefully. Reviewers need to know the difference between this paper and [2] clearly.

2. [**IMPORTANT**] Missing baseline in experiments: as I mentioned above, [2] is prior work, and it has released code. So there is no good reason for the authors not to compare GRADIENT against [2].


[1] Wang et al. Understanding Gradual Domain Adaptation: Improved Analysis, Optimal Path and Beyond. ICML 2022 (https://arxiv.org/abs/2204.08200)

[2] Klink et al. Metrics Matter: A Closer Look on Self-Paced Reinforcement Learning. 2021 (https://openreview.net/forum?id=lKcq2fe-HB)


-------------------------------------
**Post-Rebuttal**
The authors' responses addressed most of my concerns above, so I increased my rating.

---

> ### Author Response · Authors · 2022-08-02
> **Response from authors, 2/2**
>
> > Discussions on a concurrent/prior work and adding baseline WB-SPDL.
>
> ### Discussions on the differences
> We would like to expand on several significant differences in terms of the formulation and methodology between our paper and WB-SPDL [2]:
> 1. **The definition of distance function:** WB-SPDL [2] limits the W distance to be 2-Wasserstein distance under the euclidean metric (which is limited by both the formulation and algorithm implementation, see Appendix C of [2]), which is not very applicable in many cases (e.g. discrete maze as in our Figure 1). In contrast, GRADIENT utilizes the flexibility of W distance in encoding rich metric information by developing $\pi$-contextual-distance.
> 2. **The definition of Wasserstein barycenters:** WB-SPDL [2] defines the Wasserstein barycenters to be the interpolation between three distributions (Equation 6), the current distribution, target distribution, and “value distribution”. The barycenter in WB-SPDL is more of a replacement for the KL divergence in the original SPDL constrained optimization problem. Differently, GRADIENT computes directly the Wasserstein geodesics between the source and the target distribution. This simplicity in the methodology enables the theoretical analysis of why Wasserstein geodesics are proper interpolations for CRL, while [2] only provides the empirical valuation.
> 3. **The algorithm design and implementation:** the definition of barycenter in WB-SPDL [2] (which is already a difficult optimization problem) is subsequently used in the main optimization (Equation 4), which makes it a bilevel optimization problem. To circumvent this problem, WB-SPDL customizes two levels of approximation, first assuming a linear transportation map (which is only applicable in the 2-Wasserstein distance) and then assuming the W-distance can be approximated by the sum of $l_2$ distance between particles. We found that those approximations could result in computation issues (e.g., infeasible solutions during optimization) as we run their code. In their implementation, those issues require hand-crafting workarounds such as relaxing the constraint by a small factor, etc. In contrast, GRADIENT does not require such complicated approximations and directly works with off-the-shelf OT solvers.
>
> We have summarized the abovementioned differences and added them to the revised paper (Line 86-91).
>
> ### Experiment Results
> Next we added the experiments to compare GRADIENT against WB-SPDL [2] using the code in the supplementary material on OpenReview since [2] is not officially accepted by a peer-reviewed venue. We conducted a grid search on the hyperparameters $\delta, \delta_H, \epsilon$, and $\epsilon_M$ and reported the best performances. We have updated Figure 4(b) to show the results. Our method outperforms WB-SPDL in terms of asymptotic performance by 23.6% in FetchPush. By visualizing the context distribution in Figure 8(c), we found that the reason could be related to the computation issues resulting from approximations as discussed in (2) of our response to the last question. During the line search of $\beta$, sometimes the algorithm fails to find a feasible solution and jumps towards the target distribution by setting a $\beta$ close to the interval.
>
> Given the discussion and experiments, would you increase the rating If we have addressed your concerns?

---

> > ### Comment · Reviewer_G5ua · 2022-08-02
> > **Response to Authors**
> >
> > Thank you for the detailed responses, which address most of my concerns (regarding the theory & experiments). I will increase my rating to "borderline accept".

---

> > > ### Author Response · Authors · 2022-08-03
> > > **Response from Authors**
> > >
> > > We are glad our previous responses address most of your concerns regarding the theory and experiments. Thank you for raising the rating and letting us know promptly. Since your rating is borderline acceptance, I assume you will discuss the paper with other reviewers shortly to reach a consensus. We want to humbly ask whether there is any further clarification we could add to avoid confusion to readers with different backgrounds, any benchmarks we should compare with to show the pros and cons of this work, or any discussion of limitations we should add to guide future research. We are eager to provide additional evidence to facilitate your discussion and to improve the quality of this work so that it can be considered publishable.

---

> > > > ### Comment · Reviewer_G5ua · 2022-08-04
> > > > **Re:**
> > > >
> > > > I do not have more requests iso far. Thank you for all of your clarifications.

---

> > > > > ### Author Response · Authors · 2022-08-09
> > > > > **Re:**
> > > > >
> > > > > Thank you very much for the valuable suggestion and discussion! We are so glad that our clarifications served a better understanding and addressed your concerns. We deeply appreciate your support!

---

> ### Author Response · Authors · 2022-08-02
> **Response from authors, 1/2**
>
> Thank you for your review. We aim to address the concerns you raised here.
>
> > Novelty: The application of GDA and Wasserstein barycenter is quite straightforward, as long as one knows both GDA and CRL. So from the perspective of GDA, this paper is not very novel in terms of methodology.
>
> We respectfully disagree on the level of novelty of GRADIENT. We believe that GRADIENT bridges well-known techniques in GDA and CRL in a novel way. It is indeed the case that the recent works in **semi-supervised GDA** have hinted at the application of optimal transport as we pointed out in the related work section [Line 81-101]. We also notice there is a very recent work [3] that leverages Wasserstein geodesic in gradual domain adaptation for supervised learning. Similar to [1], this work also mainly focus on the distribution shift on covariates (X). We will also include this work in our related works with corresponding discussions.
>
> However, there is very little empirical evaluation or theoretical understanding of GDA in the **Reinforcement Learning** domain. Different from the semi-supervised GDA which mostly relies on the **pseudo-labeling** of existing intermediate images, CRL usually does not assume the existence of intermediate tasks and needs to generate the tasks. Although a few existing works in semi-supervised GDA resort to optimal transport when the intermediate tasks are absent as discussed in our related work section [Line 94-99], the idea has not been closely examed in previous CRL literature. Although CRL methods have achieved great empirical successes in recent years, more principled and insightful methods are still in great need. We believe GRADIENT could open the door for more analysis and understanding of CRL.
>
> As the reviewer pointed out in Strength 2, it turns out that our Theorem 1 looks similar to Lemma 1 in [1]. We would like to highlight that reinforcement learning is a different problem setting from supervised learning in [1], and the results are therefore not directly comparable. Although the goal in our Theorem 4.1 and Lemma 1 in [1] are similar, i.e., to control the loss conditioned on different distributions by the Wasserstein distance, the assumptions and all the key techniques to achieve Theorem 4.1 are totally different from [1]. We list some key points of difference here:
> 1. **Difference in the problem settings:** since RL is a sequential decision-making problem, controlling the performance gap in RL (the left-hand side of our Theorem) is more challenging. The value function in RL needs to be characterized time-sequentially and determined by a sequence of outputs depending on each other, while the loss of supervised learning is only based on one output and the label. So we believe that controlling the value function gap in RL by Wasserstein distance is novel and interesting in RL and not captured by supervised learning.
> 2. **Difference in the techniques:** 1)  Converting the performance gap to the Wasserstein distance between the probability distribution in our Theorem 4.1 is established by using the Kantorovich dual, mentioned at the beginning of Appendix A.3. While supervised learning does not require such transferring. In addition, to achieve the value function gap, we introduce some new techniques such as random walk to address the challenges in sequential decision-making problems which do not exist in prior supervised learning.
> 3. **Difference in the assumptions:** Lemma 1 of [1] relies on the Lipschitz assumptions of the classifier and the loss function, while we only need some simple bounded reward function. We achieve theoretical results mainly based on using the problem structure in RL and need great effort to arrive at the results.
>
> As the NeurIPS review guidelines (https://neurips.cc/Conferences/2022/ReviewerGuidelines) indicate: "Is the work a novel combination of well-known techniques? (This can be valuable!)", we are glad that our work provides a straightforward insight for people with strong backgrounds in domain shift in supervised learning; we also hope our work could further shed light on domain shift problems for the reinforcement learning community.
>
> **References**:
>
> [1] Haoxiang Wang, Bo Li, and Han Zhao. Understanding gradual domain adaptation: Improved analysis, optimal path and beyond. ICML 2022.
>
> [2] Kumar, Ananya, Tengyu Ma, and Percy Liang. "Understanding self-training for gradual domain adaptation." International Conference on Machine Learning. PMLR, 2020.
>
> [3] Yifei He, Haoxiang Wang, and Han Zhao. Generative Gradual Domain Adaptation with Optimal Transport. PODS workshop, ICML 2022

---

### Author Response · Authors · 2022-08-02
**General response from authors**

Thank you all for the reviews! We look forward to having constructive discussions with you in the next few days. We summarize the changes we made to the paper as follows:
* Added WB-SPDL and HER as baselines.
* Expanded discussion on related works, and clarified our novelty and contributions.
* Improved the presentation of the proof in the appendix.
* Fixed a few typos in the paper.

---

### Meta-Review · Area_Chair_7axF · 2022-08-31

**Recommendation:** Accept
**Confidence:** Less certain

**Metareview:**

The submission makes a strong case  for using optimal transport-based curriculum learning method principled theoretical insights and convincing experimental study.  At the same the there is a significant overlap between this paper and prior work on using optimal transport for curriculum learning and self-paced learning which needs to be recognized and discussed more extensively in the related work.

**Award:**

No

---

### Decision · Program_Chairs · 2022-09-14

Accept